# Interrogation of RNA-protein interaction dynamics in bacterial growth

Mie Monti[1], Reyme Herman[2], Leonardo Mancini[3,4], Charlotte Capitanchik[5,6], Karen Davey[5,6], Charlotte S Dawson [7], Jernej Ule [5,6], Gavin H Thomas[2], Anne E Willis [1✉], Kathryn S Lilley [7✉] & Eneko Villanueva [7✉]

## Abstract

**Characterising RNA–protein interaction dynamics is fundamental to understand how bacteria respond to their environment. In this study, we have analysed the dynamics of 91% of the *Escherichia coli* expressed proteome and the RNA-interaction properties of 271 RNA-binding proteins (RBPs) at different growth phases. We find that 68% of RBPs differentially bind RNA across growth phases and characterise 17 previously unannotated proteins as bacterial RBPs including YfiF, a ncRNA-binding protein. While these new RBPs are mostly present in Proteobacteria, two of them are orthologs of human mitochondrial proteins associated with rare metabolic disorders. Moreover, we reveal novel RBP functions for proteins such as the chaperone HtpG, a new stationary phase tRNA-binding protein. For the first time, the dynamics of the bacterial RBPome have been interrogated, showcasing how this approach can reveal the function of uncharacterised proteins and identify critical RNA–protein interactions for cell growth which could inform new antimicrobial therapies.**

**Keywords** *E. coli*; Proteomics; RBPome; Functional Screening; iCLIP
**Subject Categories** Microbiology, Virology & Host Pathogen Interaction; Proteomics; RNA Biology

## Introduction

The dynamic interaction between RNA and RNA-binding proteins (RBPs) is required to regulate a broad spectrum of bacterial functions, ranging from the concerted modulation of translation by Hfq (Chao et al, 2012) to the control of metabolic functions at different growth phases by CsrA (Potts et al, 2017). However, system-wide analyses of RNA–protein interactions in prokaryotes have been limited by a lack of techniques to interrogate the bacterial RBPome. Recently, specific bacteria have been engineered to polyadenylate part of their transcriptome to allow the purification of RBPs through oligo(d)T-based methods (Stenum et al, 2023). However, this approach does not allow the comprehensive evaluation of RBPs in bacteria. Alternative orthogonal approaches, either by organic extraction (Queiroz et al, 2019), silica enrichment (Shchepachev et al, 2019) and/or glycerol sedimentation (Smirnov et al, 2016) have now been successfully applied to systematically catalogue the RBPome of both gram-negative and positive bacteria (Queiroz et al, 2019; Chu et al, 2022). These studies have greatly expanded not only the number of RBPs in bacteria (ranging from ~300 to 1000 proteins)(Queiroz et al, 2019; Shchepachev et al, 2019) but also revealed unconventional RNA-binding domains in helix–turn–helix and Rossmann-fold containing proteins (Chu et al, 2022) and led to the discovery of key regulatory RBPs such as ProQ (Smirnov et al, 2016). Altogether, recent work has demonstrated that the number of RNA–protein interactions in prokaryotes had been severely underestimated.

In recent years, studying the dynamic response of the RBPome has proven fruitful in the eukaryotic context. For example, in *Drosophila* the underlying changes in the RBPome upon maternal to zygotic transition were investigated using an oligo(d)T-based approach (Sysoev et al, 2016). Hundreds of high-confidence proteins were identified, and half were not previously known to interact with RNA. Similarly in macrophages, 402 proteins were found to bind RNA upon stimulation with lipopolysaccharides, several of which exhibiting biochemical functions not directly related to RNA interaction, such as P23, a co-chaperone of HSP90, was further demonstrated to interact with mRNAs (Liepelt et al, 2016). In breast carcinoma cells, DNA damage was found to increase polyA-RNA binding for 260 proteins including novel regulators of splicing efficacy promoting survival after ionising radiation (Milek et al, 2017). These studies suggest a key role for many previously uncharacterised RBPs in coordinating dynamic biological responses. However, no studies to date have examined how bacterial RBP-RNA interactions change following perturbation.

In this study, we have applied Orthogonal Organic Phase Separation (OOPS), a method to retrieve RBPs that is agnostic to RNA biotypes (Queiroz et al, 2019; Villanueva et al, 2020), to

[1]MRC Toxicology Unit, University of Cambridge, University of Cambridge, CB2 1QR Cambridge, UK. [2]Department of Biology, University of York, Wentworth Way, York YO10 5DD, UK. [3]Cavendish Laboratory, University of Cambridge, Cambridge CB3 0HE, UK. [4]Department of Biochemistry, University of Cambridge, Cambridge CB2 1QW, UK. [5]The Francis Crick Institute, 1 Midland Rd, London NW1 1AT, UK. [6]UK Dementia Research Institute at King's College London, The Wohl, 5 Cutcombe Road, London SE5 9RX, UK. [7]Cambridge Centre for Proteomics, Department of Biochemistry, University of Cambridge, CB2 1QR Cambridge, UK. ✉E-mail: aew80@mrc-tox.cam.ac.uk; k.s.lilley@bioc.cam.ac.uk; ev318@cam.ac.uk

characterise how the RBPome is dynamically rewired during different phases of *E. coli* cell growth. Our work reveals the RNA-binding dynamics of 271 RBPs. RNA-binding activity has been detected for 17 proteins with previously unannotated function and five of these RBPs have been found to be required for efficient bacterial growth. We have further characterised the interactome of YfiF, the RBP whose removal has the greatest impact on *E. coli* growth. YfiF, a protein with a methyltransferase domain, not only interacts with rRNA and tRNAs, but also with the regulatory non-coding RNAs (ncRNAs) encoded by *csrB* and *arrS*, as well as with its own RNA, suggesting that the function of this protein may be riboregulated. Our work further unveils unexplored RBP function of highly evolutionary conserved proteins, including two proteins with human mitochondrial orthologs associated with metabolic rare disease and suggests a functional specialisation of RBPs from the bacterial to the eukaryotic organelle level. Finally, we determine alternative RBP functions for proteins with well-characterised functions not related with RNA binding such as HtpG, a known bacterial protein chaperone involved in extraintestinal pathogenic *E. coli* virulence (Garcie et al, 2016). We characterise HtpG as a tRNA binder with increased RNA binding in the stationary phase. Notably, HtpG interactors include LeuX, a suppressor tRNA implicated in the expression of pathogenic factors (Dobrindt et al, 2002).

In conclusion, we present the first dynamic RBPome of a bacterium, the model organism *E. coli*, which reveals extensive reorganisation of RNA–protein interactions during cell growth stages. Importantly, this work also showcases how the interrogation of dynamic RNA and protein interactions in prokaryotes can highlight new key players in cell physiology as well as potential new antimicrobial targets.

# Results

## Exploring the *E. coli* proteome at different cell growth stages

We first evaluated *E. coli* protein dynamics along the bacterial growth stages assessing how the proteome adapts during lag, exponential and stationary phases of growth in batch culture (Fig. 1A). Overall, changes in the abundance of 2360 proteins were quantified in all three phases, representing ~91% of the estimated expressed *E. coli* proteome (Soufi et al, 2015). While different growth stages display unique proteome signatures (Fig. 1B), further analyses of the variation in protein abundance in the different phases by linear modelling revealed six distinct profiles (Fig. 1C,D; Dataset EV1). As expected, proteins upregulated in the exponential growth phase (Groups 2 and 3) are implicated in metabolic processes which are required for cellular adaptation to robust growth (Fig. 1E). Conversely, homoeostatic processes are enriched in the stationary phase (Group 4, Appendix Fig. S1). Interestingly, GO-term enrichment analysis of the proteins of the different groups indicated that proteins implicated in RNA homoeostases such as ribosome biogenesis and translation, are among the most differentially expressed proteins across the different bacterial growth phases, suggesting that RBPs may play a central role during the transition between exponential and stationary growth phases (Fig. 1E; Appendix Fig. S1; Dataset EV2). These data suggest a central role for RBPs in bacterial growth control.

## Capturing RNA-binding dynamics

To quantify the RNA-binding capability of RBPs at different stages of population growth, we started by comprehensively characterising the *E. coli* RBPome using OOPS (Queiroz et al, 2019). To do so, we cross-linked RNA–protein interactions using UV light at 254 nm and RNA–protein adducts at acidic phenol-chloroform interfaces. Traditional protocols requiring UV-cross-linking require removing the media where the cells are grown (e.g., LB medium) to allow the UV to reach the cells. This may result in quick adaptation to the new medium, potentially altering biologically relevant RNA–protein interactions. To avoid this, cells were cultured in M9 glucose medium supplemented with 10% LB (M9LB) medium, enabling us to directly cross-link in the culture medium thus avoiding additional washes which may produce spurious biological signals. The *E. coli* RBPome was defined as the set of proteins sensitive to RNase treatment in OOPS (Villanueva et al, 2020) (Appendix Fig. S2a; Dataset EV3). Previously identified RBPs were confirmed using this method (Appendix Fig. S2b), whereas other nucleic acid-binding proteins, such as the single-stranded DNA binding protein ssDBP were not sensitive to RNases and therefore are not included in the RBPome (Appendix Fig. S2c). Using this approach, 271 proteins were classified as RBPs in *E. coli*. This RNA-agnostic method greatly expands the 169 RNA-interacting proteins in *E. coli* obtained using RIC (Stenum et al, 2023). Furthermore, introducing an RNase sensitivity filter to classify RBPs further refines previous RBP catalogues obtained by phase separation (Queiroz et al, 2019) (Appendix Fig. S2d). Importantly, 170/271 RBPs were not previously GO-annotated as RNA interactors (Appendix Fig. S2e–g). As previously described, the resulting RBPome was enriched in proteins that were significantly more hydrophobic, positively charged and enriched in K, Y, R amino acids than the overall *E. coli* proteome (Appendix Fig. S3).

We evaluated how these 271 RBPs modify their RNA-binding capacity along the lag, exponential and stationary phases. We used OOPS to obtain, both the total proteome and the RBPome, from the same sample, and quantified the differences in protein abundance by Tandem Mass Tag (TMT) Mass Spectrometry (Fig. 2A). Quantifying total protein abundance and RBP abundance from the same sample allowed classification of RBPs in three main groups which: (i) do not change abundance or RNA-binding across the phases; (ii) change RNA-binding according to their total protein abundance; or (iii) change their RNA-binding independently to their abundance (Fig. 2B; Dataset EV4). We found that 184 proteins belong to clusters ii or iii meaning that 68% of the bacterial RBPome differentially binds RNA according to the bacterial growth stage.

Of the RBPs that change their RNA binding between the exponential and stationary growth phases, we observed that the RNA binding of 64% of the bacterial RBPome correlates with general protein abundance (quadrants 1 and 3 in Fig. 2C). Consistent with the decrease of translation in stationary phase and the fact that over half of the GO-annotated RBPs are components of the translational machinery (Holmqvist and Vogel, 2018) 60% of the GO-annotated (Chi-square test, *P* value < 0.0001) proteins with a role in protein synthesis decreased in both total abundance and RNA binding (quadrant 3 (Q3), Fig. 2C). This includes CspA, a cold-shock protein known to decrease its abundance and activity after the early-log phase (Jones et al, 1987; Goldstein et al, 1990) (Fig. 2D). Q3 also contains 16 30 S

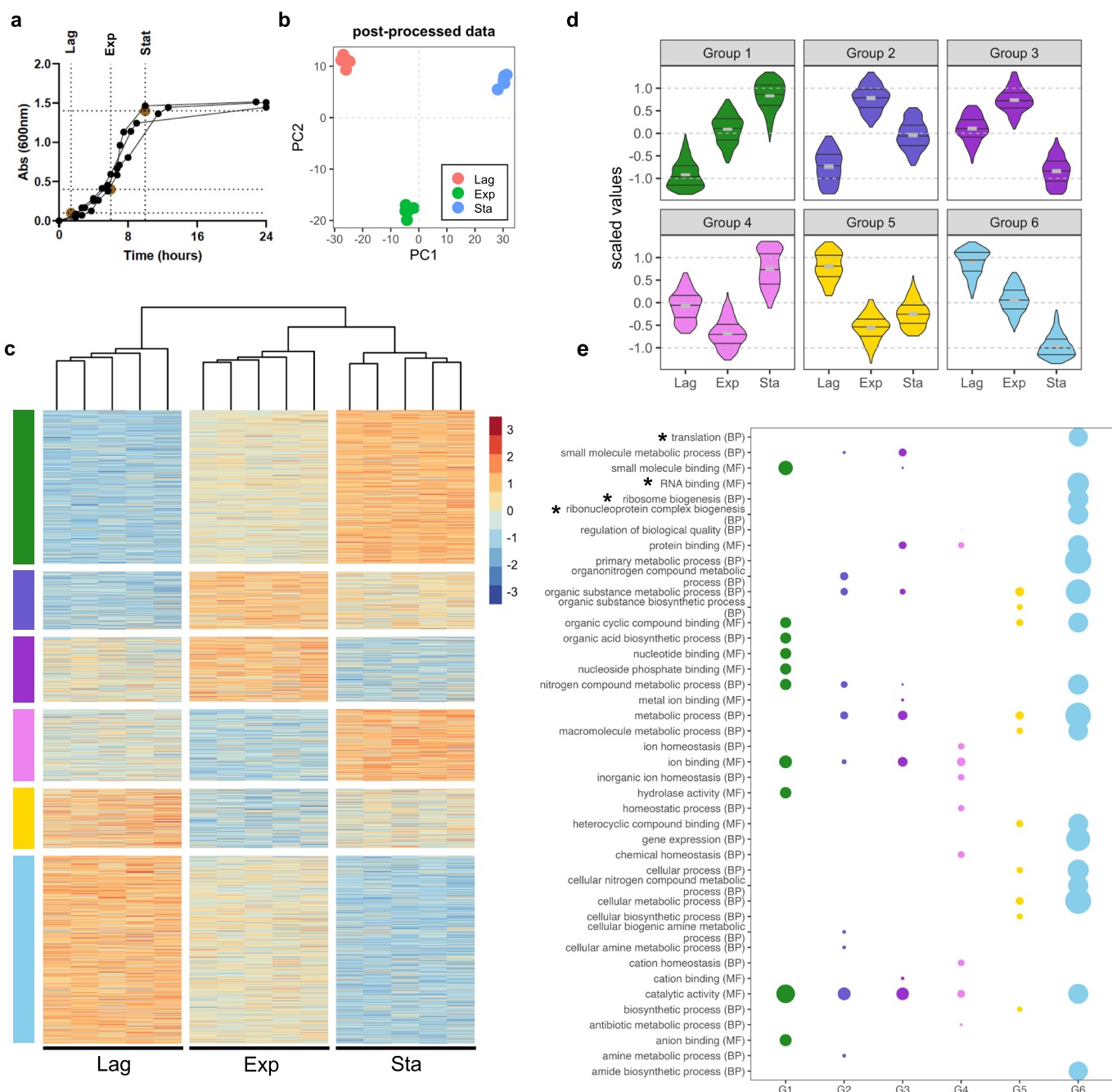

**Figure 1. *E. coli* growth curve dynamic proteome.**

(A) *E. coli* growth curve indicating sampling points for lag, exponential and stationary growth phase. Three independent replicates shown. (B) Principal component analysis (PCA) of protein abundances. (C) Protein abundance values, abundance *z*-scores normalised across all samples. Samples hierarchically clustered across all extractions as shown above by Pearson correlation as the distance metric. Proteins grouped by linear modelling as shown by colour bars on the left. Five independent replicates per growth phase. (D) Violin plots of protein abundance profiles for each defined group. Group 1 ($n = 610$), Group 2 ($n = 229$), Group 3 ($n = 257$), Group 4 ($n = 284$), Group 5 ($n = 235$), and Group 6 ($n = 745$). (E) GO-term enrichment of top terms for each profile, coloured as shown in (D). Size is inversely proportional to *P* value (Fisher's Exact Test with Bonferroni correction for multiple testing). In bold, terms directly relevant to RNA binding. Asterisk (*) highlighting RNA biology-related terms.

ribosomal proteins (out of 17 identified) and 22 50 S ribosomal proteins (out of 25 identified), along with other proteins implicated in canonical RNA processes (Fig. 2E). Only 5% of the GO-annotated RBPs increase in abundance and RNA binding at the stationary phase (Q1), including Rmf (Q1, Fig. 2C,D), a ribosomal

hibernation factor known to increase its activity in the transition from exponential to stationary, when it reversibly converts 70 S ribosomes to their inactive dimeric form (Wada et al, 1995, 1990). The remainder of proteins in Q1 have not been annotated as RNA modulators and their most significant GO-term was 'catalytic

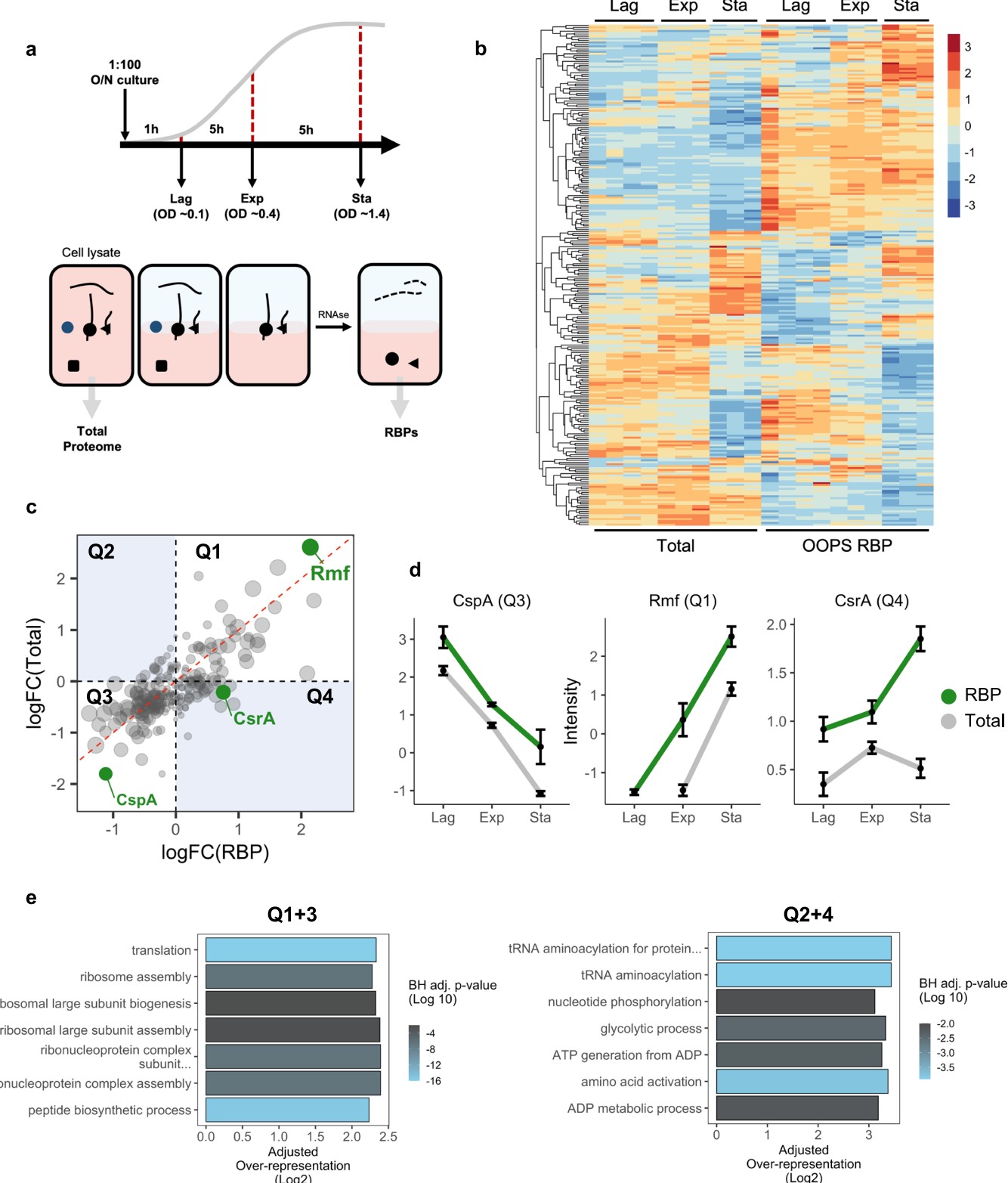

activity' (Fig. 2E; Appendix Fig. S1). Many eukaryotic proteins with catalytic activity e.g., glycolytic enzymes, have the capacity to bind nucleotides as part of their enzymatic role and are classified as 'moonlighting RBPs' (Castello et al, 2015; Hentze et al, 2018; Cieśla, 2006; Curtis and Jeffery, 2021). Indeed, over 35% of the proteins

annotated as having catalytic activity in group 1 are also annotated as nucleotide binders. These results suggest the role of nucleic acid-binding proteins as RBPs could be conserved through evolution.

Of particular interest were proteins whose RNA binding did not correlate with protein abundance (Q2 and 4, Fig. 2C). This non-linear

**Figure 2. RNA–protein interactions across *E. coli* growth.**

(A) Top: schematic representation of growth curve sampling strategy. Cells were cross-linked and harvested at lag, exponential and stationary phase. Bottom: schematic representation of protein extraction. Total proteome extracted from cell lysates and RBPs extracted following OOPS procedure. (B) Protein abundance from total proteome and RBPome extractions. Abundance z-score normalised across all samples. Proteins are hierarchically clustered across all samples as shown on the left. Four independent samples for lag phase, three for log and stationary phase for total and RBPome. (C) Quadrant plot of total protein and RBP abundance fold changes between exponential and stationary phases. Rmf, CspA and CsrA highlighted in green. Size of data point is inversely proportional to *P* values. Statistical analysis performed using linear modelling followed by moderated *t* tests and empirical Bayes method to assess differential expression (see 'Methods'). (D) Protein abundance levels for CspA, Rmf and CsrA, data shown as mean across all samples and extraction types. Error bars represent standard error ($n = 3$ independent replicates). (E) GO-term over-representation analysis for quadrants 1 and 3 (Q1 + 3) and quadrants 2 and 4 (Q2 + 4), against all proteins identified. BH adj. *P* value is the Benjamini–Hochberg adjusted *P* value obtained from modified hypergeometric test to account for protein abundance.

behaviour suggests that the RNA-binding capacity of those proteins could be regulated by alternative mechanisms to protein abundance. Out of the 66 proteins where RNA binding was not correlated to protein amount between stationary and exponential phases (FDR <0.01), only 24 were annotated as RBPs. This includes CsrA, a bacterial RBP responsible for binding and repressing the expression of glycogen-synthesis genes during the stationary phase (Potts et al, 2017; Baker et al, 2002; Romeo et al, 1993). In keeping with these findings, we found a significant increase in RNA-binding of CsrA (*P* value: 1.92E-7) in the stationary phase, that is independent of the total abundance of CsrA protein (Fig. 2D). This reinforces that integrating total protein abundance and RNA-binding capacity allows capture of complex regulatory RNA-binding dynamics (Sysoev et al, 2016; Villanueva et al, 2020; Perez-Perri et al, 2023).

## YfiF as a novel RBP fundamental for bacterial growth

We then focused on the 17 proteins, ~6% of the RBPome, that were poorly characterised and lacked functional annotations. Within this subset, 8 RBPs were shown to significantly change their RNA-binding activity between the stationary and exponential phases (Fig. 3A). Three of these uncharacterised proteins have been suggested to interact with the translation machinery in interaction-screenings: YfiF physically interacts with rRNA modifier proteins (GeZi et al, 2021); YbcJ associates with the 50 S ribosomal subunit (Jiang et al, 2007); and PhoL (YbeZ) is known to interact with YbeY, a ribosome quality control and 16 S maturation factor (Vercruysse et al, 2016). Finally, although YebC, has been suggested as a putative transcriptional regulator (Skunca et al, 2013), previous work has classed it as a potential RNA interactor in *Staphylococcus aureus* (Chu et al, 2022).

To explore the role of the eight unannotated dynamically regulated-RBP in bacterial growth, we analysed the effects of knocking out these proteins and evaluated the growth phenotype and morphology in rich and growth-limiting defined media (Fig. 3B; Appendix Fig. S4). Four of the candidates displayed significantly decreased growth rates and lower final growth yield, with the *yfiF* deletion strain having the strongest phenotype, supporting recent observations of its importance for bacterial growth (GeZi et al, 2021). While not previously characterised as an RBP, YfiF has a predicted RNA-binding domain within its structure (Gaudet et al, 2011) (Fig. 3C). Moreover, it has been found to physically interact with previously characterised RBPs including Rne, Rnr and RluC (Fig. 3D,E; Appendix Fig. S5a), and PhoL, another unannotated RBP with the second strongest growth phenotype upon its knockout (Fig. 3B,D). Intriguingly, a YfiF KO did not show changes in cell morphology over the growth curve (Appendix Fig. S4), in disagreement with previous work (GeZi et al, 2021). To clarify the role of YfiF

as a novel RBP, we further characterised its interactome using iCLIP (Appendix Fig. S5b–e). In agreement with its postulated role as a ribosomal dormancy factor and putative tRNA methyltransferase, we find that YfiF interacts with both rRNAs and tRNAs in vivo (Fig. 3F,G; Appendix Fig. S5f). The top RNA motif was **AACCTTTACW** (Fig. 3H), which was identified in 16 peaks located at rRNAs (Dataset EV5). Surprisingly, YfiF also showed consistent binding to non-coding RNAs (Fig. 3G; Appendix Fig. S6), including csrB/C, arrS and chiX, all of which have a role in cell survival in acidic conditions (Aiso et al, 2014; Babitzke and Romeo, 2007; Hayes et al, 2006). As the cells were cultured in M9LB which contains glucose, we expect fermentation and acidification of the media throughout batch culture (Walczak et al, 2023), suggesting a putative upregulation of these ncRNAs. Positionally enriched k-mer analysis (PEKA) (Kuret et al, 2022) revealed a preference for AU-rich k-mers seen in the cases of ncRNAs (Fig. 3I; Appendix Fig. S7). Importantly, YfiF also interacts with its own RNA (Fig. 3J), a common feature of several RBPs (Wolin et al, 2023) that has been previously described as a mechanism to control their own mRNA expression (Hu et al, 2014).

## Evolutionary conservation of uncharacterised RBPs

To evaluate the level of conservation of the 17 new RBPs in *E. coli* which lacked functional annotation, we performed a systematic conservation analysis. As expected, the number of orthologs of these proteins decreases with evolutionary distance (Fig. 4A), with Proteobacteria having the highest number of orthologs in common. Proteobacteria include a significant number of human pathogens from the *Salmonella*, *Yersinia*, *Vibrio*, *Bordetella* or *Brucella* genus. Interestingly, we found that while some RBPs such as YebC or YjjV are conserved across almost the entirety of the phylum, some RBPs such as YbcJ, YdhQ or YjpA are almost exclusively present in pathogenic species (Fig. 4B; Dataset EV6).

Remarkably, we found three RBPs with conserved orthologs between *E. coli* and *H. sapiens*: YhgF, YgfZ and YebC, which are nuclear-encoded RBPs localising to the mitochondria. Interestingly, YebC (TACO1) and YhgF (SRBD1) genes trace back to the last universal common ancestor, while YgfZ (IBA57) is mostly present in the proteobacteria family, suggesting that the ancestral IBA57 gene may have been domesticated. Significantly for YgfZ, this gene is retained in the reduced genomes of related bacterial symbionts of insects (Prickett et al, 2006), suggesting its function has become essential in these environments. Importantly, the human orthologs of these bacterial RBPs have been found to interact with RNA (Queiroz et al, 2019; Baltz et al, 2012; Castello et al, 2016). Both YhfG and YgfZ have conserved RNA-binding domains, while YebC has a highly conserved HTH domain recently found to bind RNA

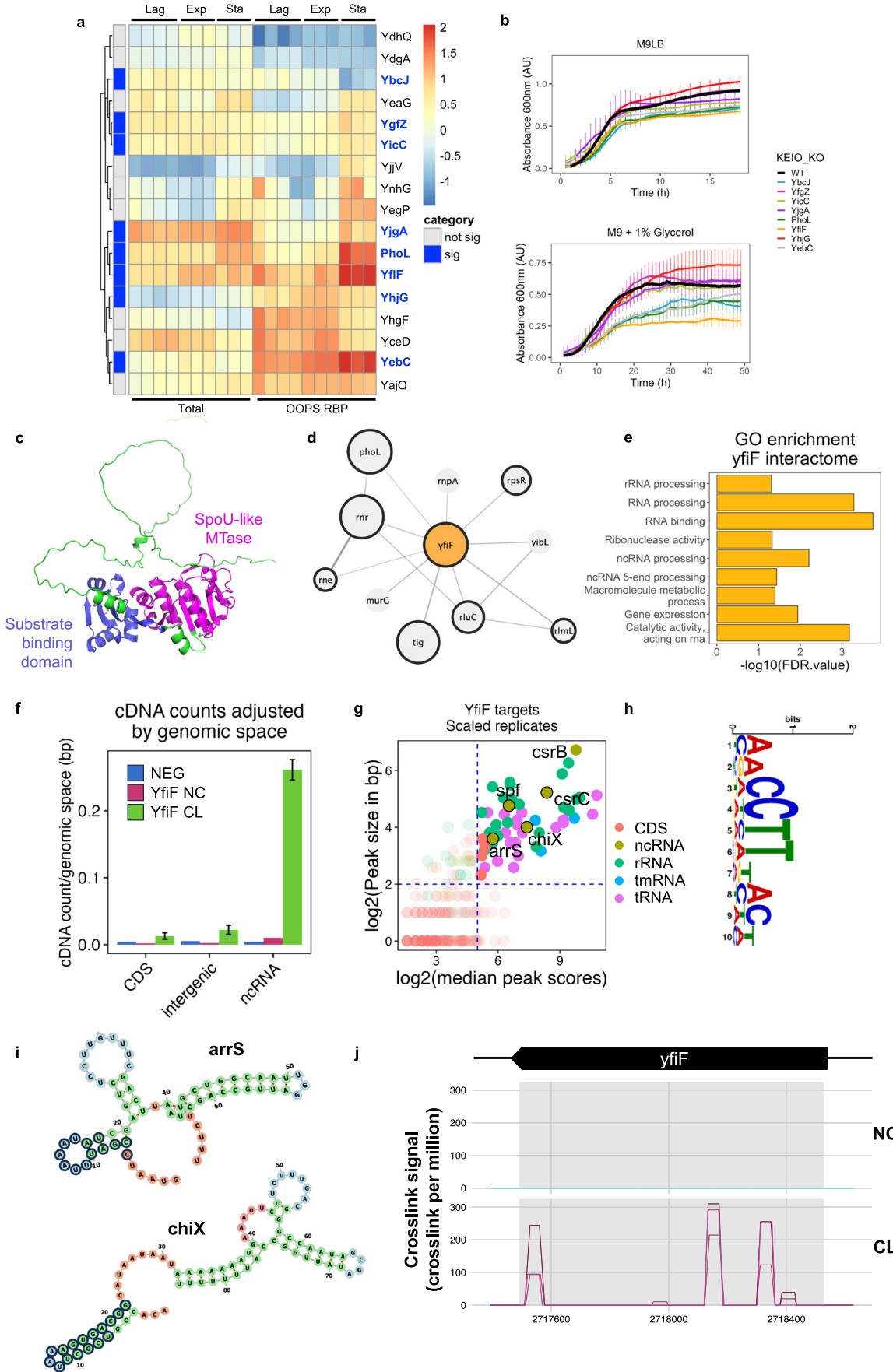

**Figure 3. Characterisation of Yfif as an RNA-binding protein.**

(A) Unannotated RBP protein abundance and RNA-binding dynamics. Abundance *z*-score normalised within each extraction type. Proteins are hierarchically clustered across all samples as shown on the left. Proteins highlighted in blue are those which significantly change RNA-binding between the exponential and stationary phase. (B) Knockout growth curve with standard deviation as error bars in M9LB (top) and M9 + 1% glycerol (bottom) growth media of all proteins with significant changes in RNA binding in (A). Three independent replicates per line. Data shown as mean values. Error bars represent standard deviation ($n = 3$ independent replicates). (C) Alphafold structure prediction of YfiF structure (AF-P0AGJ5-F1). Protein shown as green surface. Canonical RNA-binding domains highlighted labelled in blue and magenta. (D) Physical interaction network for YfiF as predicted by STRING-db (Szklarczyk et al, 2021). Proteins with black borders are RBPs; size of node inversely proportional to *P* value of dynamic RNA-binding between exponential and stationary phase retrieved by linear modelling analysis with the limma R package followed by moderated *t* tests. (E) GO-term enrichment of YfiF-predicted interactome. (F) Unique cDNA molecules cross-linked to YfiF protein and aligned to *E. coli* genome, values adjusted over total genomic space of the biotype. Reads aligned to tRNA and rRNA located in the 'intergenic' category. Data shown as mean values. Error bars represent standard deviation ($n = 3$ independent replicates). (G) Peaks overlapping in all three replicates and not observed in either control were plotted by median score and size. Highlighted are targets with a log2(median peak score) >5 and a log2(peak size) >2. Labelled are the ncRNA passing these threshold values. (H) Top motif calculated by STREME using YfiF NC peaks as the control sample. (I) RNA structures of two ncRNA YfiF targets as predicted by ViennaRNA (Version 2.6.3) minimum fold energy structure. Highlighted in black circles is the binding site as determined by iCLIP. (J) Analysis of iCLIP datasets mapped to the *yfif* gene. Cross-link counts are visualised and normalised to library size with CLIPplotR. Three independent replicates per CL sample were performed with one replicate devoted to each non-cross-linked sample.

in vivo (Chu et al, 2022) (Fig. 4C,D; Appendix Fig. S8a–d). While annotations for YhgF are very limited and the function of the human ortholog (SRBD1) is unclear, the RNA interactome of this protein has been recently characterised in bacteria where it was found to bind both mRNAs, tRNAs and sRNAs. Indeed, over-expression of YhgF leads to increased levels of its target *rmf* mRNA indicating a role in transcriptional regulation (Stenum et al, 2023). Interestingly the two other evolutionarily conserved RBPs have been associated with a number of rare diseases in humans. The mitochondrial YgfZ ortholog (IBA57) is implicated in the synthesis of Fe/S proteins and mutations in the IBA57 gene have been linked with hereditary spastic paraplegia (Lossos et al, 2015) and a metabolic syndrome presenting with severe myopathy and encephalopathy(Ajit Bolar et al, 2013). While its RNA-binding partners have yet to be characterised, IBA57 has been identified as an RBP in high-throughput assays (Baltz et al, 2012). Finally, the human ortholog of YebC (TACO1) encodes a translational activator of COXI (Queiroz et al, 2019; Castello et al, 2016). Defects in TACO1 result in cytochrome C oxidase deficiency and late-onset Leigh syndrome (Weraarpachai et al, 2009). Of note, we found that YebC not only binds RNA differentially at different stages of bacterial growth, but its knockout displays a severe growth defect phenotype, suggesting that its role as an RBP and as a putative translational modulator could be conserved from bacteria to human. Taken together, these results point to a differential conservation of the newly detected RBPs, showing proteins exclusively conserved in Proteobacteria and others showing conservation across kingdoms. Proteobacteria-specific RBPs, and especially those enriched in pathogenic microbes with the capacity to control cell growth, could be explored as new antibiotic drug targets. Alternatively, the highly conserved proteins described by this study, open up new questions on the functional diversification of RBP function across evolution (Appendix Fig. S5c).

## Protein chaperones as bacterial RBPs

In addition to those proteins without previously annotated functions, we found that three of the RBPs with the most significant changes between exponential and stationary phases have been previously described as protein chaperones, and none are currently annotated as RNA binders (Fig. 5A; Appendix Fig. S6a). All three displayed increased RNA binding in the stationary phase alongside a concomitant increase in protein abundance (Q1, Fig. 2C), showing consistent binding profiles across the

phases (Fig. 5B). Interestingly, all three proteins are part of the same protein complex as annotated in the STRING database (Szklarczyk et al, 2021) (Fig. 5C; Appendix Fig. S6b,c). Moreover, HtpG has been recently reported to require the direct binding and collaboration of DnaK to function (Corteggiani et al, 2022). While not annotated as RBPs in bacteria, both DnaK and HtpG are conserved from bacteria to humans, and their human orthologs have been consistently identified as RBPs (Queiroz et al, 2019; Baltz et al, 2012; Castello et al, 2012). Importantly, the RNA-binding peptides identified in the human orthologs of both DnaK and HtpG are highly conserved in *E. coli* (Fig. 5D; Appendix Fig. S9d). While a direct ortholog of DnaK has been found to bind tRNA in humans(Leone et al, 2023), RNA interactors of HtpG are still unknown. The RNA-binding of HtpG was confirmed via a PNK assay (Extended Data S10). To identify the RNAs that interact with HtpG in *E. coli*, iCLIP analysis was carried out (Huppertz et al, 2014) (Appendix Fig. S7a–d) as applied with the previous candidate. Peaks overlapping in both experimental replicates and not in the controls were taken forward and visualised by peak score and size, showing a preference for ncRNAs genes, mostly tRNAs (Fig. 5E,F; Extended Data S10e). To explore whether there was a particular structure or sequence motif within the tRNAs for HtpG binding, the CL peaks were analysed through STREME (Bailey, 2021) using the NC samples as a negative control. The top motif was **HGGWTTTYAA** (Fig. 5G), identified in 16 of the target tRNAs consistently in the anticodon binding loop (Fig. 5H; Dataset EV7). Furthermore, we found that HtpG interacts with the LeuX transcript (Appendix Fig. S10f,g). LeuX codes for a suppressor tRNA that inserts leucine at the amber codon and is required for the translation of virulence factors in pathogenic *E. coli* strains (Dobrindt et al, 2002). Importantly, LeuX maturation is tightly regulated in a cell growth-dependent manner (Nomura et al, 1987), correlating with HtpG increase in RNA binding (Fig. 5F; Appendix Fig. S10f). Altogether, our unbiased analysis allows assigning novel RNA-binding capabilities to bacterial proteins and uncovers the interplay between proteins and RNA species implicated in bacterial pathogenesis.

## Discussion

This work presented here represents the first dynamic character-isation of RNA–protein interaction rewiring through different stages in the population growth of bacteria. Using the model organism *E. coli*, we have queried both the proteome and the RBPome at the lag, exponential and stationary growth phases. This

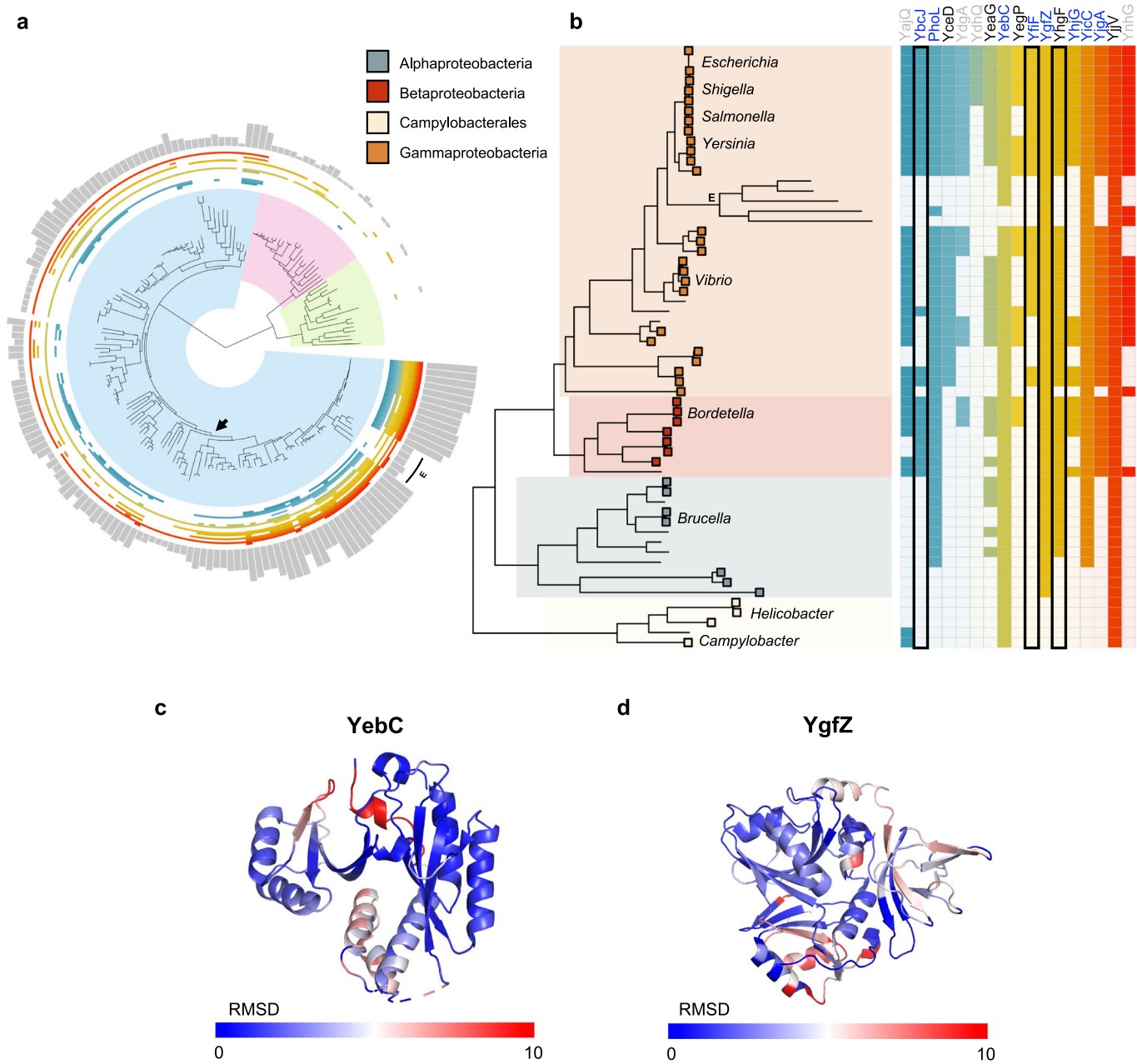

**Figure 4. Evolutionary conservation of unannotated proteins.**

(A) Tree of life covering the three Domains: bacteria (blue), archaea (pink), and eukarya (green). Heatmap wrapping the tree indicates the presence of an ortholog for each of the 17 candidate proteins. Bar plot indicates the sum of orthologs per tip (maximum value 17) Data available in Dataset EV6. Arrow indicates the branch of proteobacteria phylum. (B) Proteobacteria phylum, where the box at tip represents pathogenic species found at leaf tip. Heatmap indicates presence of ortholog, if unfilled no ortholog identified. Protein names in blue are dynamic RBPs belonging to Q2/4, in black belong to Q1/3. Protein names in grey do not show significant changes in RNA-binding across the growth curve. Black box highlighted proteins with canonical RNA-binding motif. E endosymbionts. (C) YebC (1KON) and its human ortholog (TACO1; AF-Q9BSH4-F1) aligned and coloured by RMSD. Only YebC shown here. Dark blue represents a high alignment score, higher deviations are in red. (D) YgfZ (1VLY) and its human ortholog (IBA57, 6QE3) aligned and coloured by RMSD as in (C). Residues not used for alignment are coloured grey.

has revealed that 77% of bacterial RBPs change their abundance (FDR < 1%) along the different growth stages and that 68% modify their RNA interaction between exponential and stationary states (FDR < 1%). Our analyses unravel new RBP functions for previously annotated bacterial proteins, such as the HtpG chaperone. Importantly, they also uncover the RNA-interaction

properties of 17 experimentally unannotated *E. coli* proteins. This has led to the characterisation of YfiF as a tRNA-binding protein as well as a protein required for bacterial growth.

Comprehensive RBP characterisation in bacteria is under-explored in comparison with eukaryotes, mainly due to previous technical limitations. Here, we have taken advantage of our

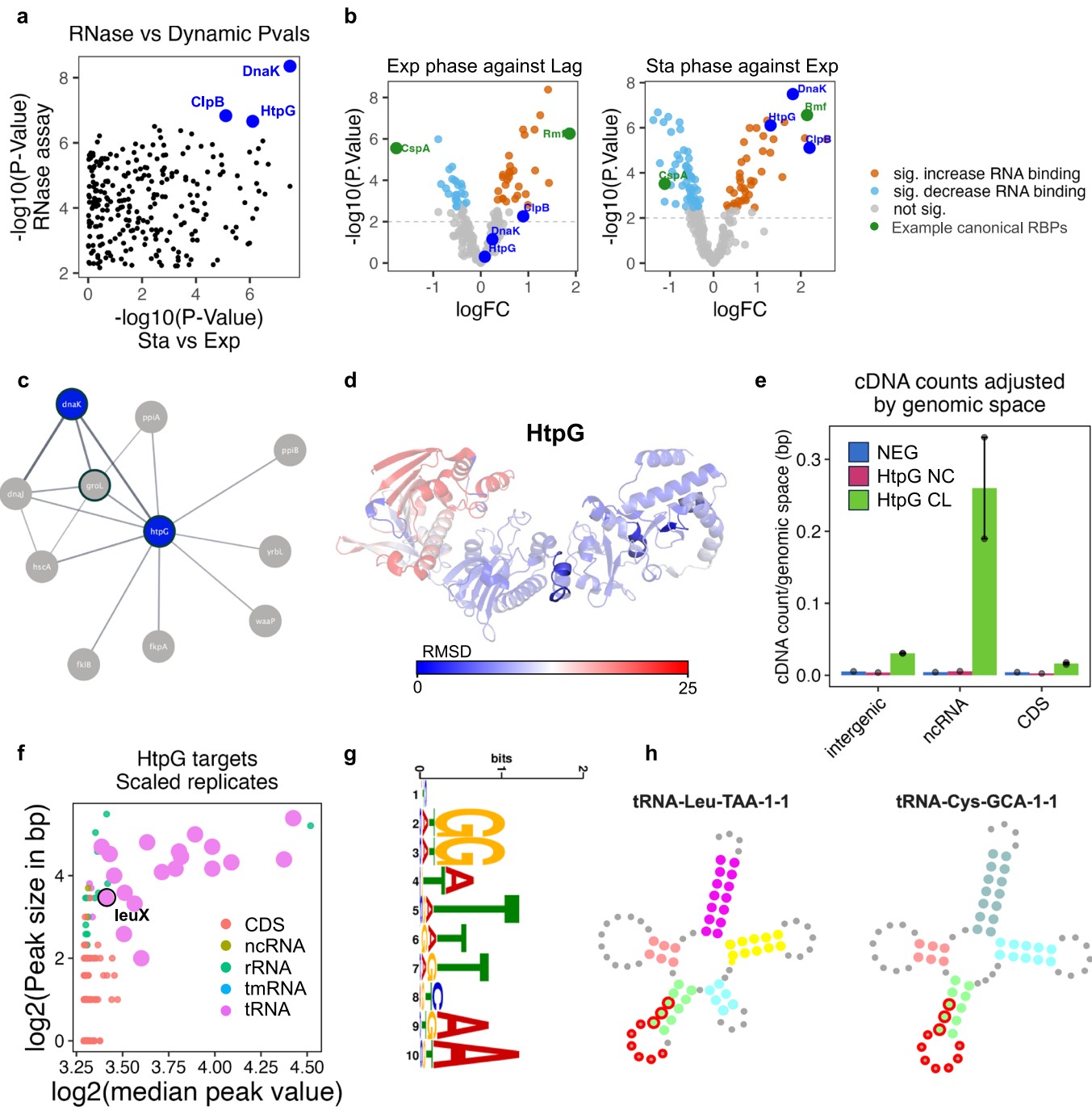

**Figure 5. HtpG characterisation as *E. coli* RBP.**

(**A**) Correlation between the *P* values of the differences in RNA-binding dynamics between the stationary and exponential phases, and the RNase assay. Statistical analysis performed using linear modelling followed by moderated *t* tests. (**B**) Volcano plot representations of RBPs between the different growth phases. CspA and Rmf (canonical RBPs) highlighted in green, molecular chaperones in dark blue. Statistical analysis performed using linear modelling followed by moderated t tests and empirical Bayes method to assess differential expression (see 'Methods'). (**C**) Physical interaction network of HtpG as predicted by STRING-db. In blue, candidates highlighted in (**A**, **B**). Proteins detected as RBPs outlined in bold. (**D**) Alphafold structure prediction of entire HtpG sequence (AF-P0A6Z3-F1) coloured by RMSD score when aligned with human ortholog (HSP90AA1). In opaque, the peptides are predicted to bind RNA in the human ortholog as predicted by (Queiroz et al, 2019; Castello et al, 2016), all found in the structurally conserved C-terminal domain. (**E**) Unique cDNA molecules cross-linked to HtpG protein and aligned to *E. coli* genome, values adjusted over total genomic space of the biotype. Reads aligned to tRNA and rRNA located in the 'intergenic' category. Data shown as mean values. Error bars represent standard deviation (*n* = 2 independent replicates for CL, *n* = 1 for controls). (**F**) HtpG peaks found only in the 'HtpG CL' replicates plotted by median peak size versus peak value. tRNA genes are highlighted in violet. (**G**) Top HtpG binding motif as predicted by STREAM. (**H**) Schematic representation highlighting the identified binding peak of HtpG on the secondary structure of tRNA Leu-TAA-1-1 and Cys-GCA-1-1, delineated in red.

previously developed methodology, OOPS (Queiroz et al, 2019; Villanueva et al, 2020), to catalogue and characterise the dynamics of the *E. coli* RBPome. As previously shown, the recovery of RNA biotypes bound by RBPs via OOPS is unbiased for transcripts over 60 bp (Queiroz et al, 2019). Since OOPS requires UV-cross-linking of RNA–protein interactions, it will retrieve RBPomes constituted by those proteins that interact with RNA at less than 4 Å and are proximal to a uracil nucleotide. While some RBPs may not be amenable to UV-cross-linking, this approach allows for a stringent discrimination between direct and indirect RNA interactors in ribonucleoprotein complexes.

Our analysis of the RBPome dynamics during cell growth is consistent with targeted studies of RNA-interacting proteins in *E. coli* and other bacteria. We not only recapitulate previously established RBP dynamics, such as an increase of Rmf RNA-binding between the exponential and the stationary phase (Wada et al, 1995, 1990) or an increased binding of translation-related RBPs from the lag to the exponential phase, but also more complex regulatory profiles. This includes the increase in RNA binding of CsrA irrespective of its abundance during cell growth. Importantly we have identified 17 partially or fully uncharacterised RBPs which dynamically change their RNA binding in the stationary phase. The case of YfiF is of particular interest: While current information about this protein is limited, it is only present in Proteobacteria, and its absence is known to have a detrimental effect in the growth of *E. coli*. Interestingly, the predicted RNA-binding domain of YfiF suggests that it could act as a ribosomal dormancy factor (GeZi et al, 2021). Here, we find that YfiF not only interacts with rRNAs and tRNAs, but also with other regulatory ncRNAs as well as with its own RNA. Given that this protein has a methyltransferase domain, we hypothesise that this RBP acts as an rRNA and tRNA methyltransferase, and that its function is regulated by the binding of other regulatory ncRNAs such as rnpB, and this will form the basis of further studies. Moreover, YfiF binds its own RNA, therefore it is possible that YfiF expression could be further regulated by a post-transcriptional feedback loop, as it has been found for other RBPs (Hu et al, 2014). Importantly, its impact on cell growth and the fact that YfiF is specific to the Proteobacteria family, indicates it could represent a new target for antibiotics. Interestingly, 3 of the unannotated bacterial RBPs have orthologs in humans. Strikingly, all of these are mitochondrial proteins and two of them (YgfZ and YebC) have been associated with mitochondrial-linked metabolic syndromes (Lossos et al, 2015; Ajit Bolar et al, 2013; Weraarpachai et al, 2009). This can suggest further studies to determine the role for these proteins across evolution and unveil the process of functional specialisation from bacterial to organelle homoeostasis.

Finally, we have also shown that protein chaperones such as HtpG bind RNA in *E. coli*, specifically at the stationary phase. HtpG is required for the synthesis of secondary metabolites implicated in extraintestinal pathogenic *E. coli* virulence (Garcie et al, 2016), while its human ortholog has a critical role in mitochondrial protein import (Young et al, 2003). In this study, we reveal that HtpG may have an extra function as a tRNA-binding protein. This is especially relevant considering that small RNA species can be underrepresented in OOPS owing to inefficient recovery during aqueous:organic phase separation (Queiroz et al, 2019). Importantly, our qualitative analysis shows that HtpG interacts with LeuX, a suppressor tRNA required for the translation of virulence

factors in pathogenic *E. coli,* suggesting an additional mechanism linking HtpG function to *E. coli* virulence.

Taken together, this study provides the most comprehensive catalogue of protein dynamics at the different stages of bacterial growth and represents the first dynamic assessment of RBP rewiring during the *E. coli* growth cycle. We showcase how the interrogation of RBP dynamics can be used to discover the functions of uncharacterised proteins in any bacteria. Therefore, we anticipate this approach will allow exploring new RBP functions, and their evolution and adaptation across kingdoms. Furthermore, characterising essential protein–RNA interactions for pathogenic bacterial survival could reveal novel targets for the development of antimicrobials.

# Methods

## Bacterial growth media M9LB

Media prepared with autoclaved 15.6 g M9 minimal media salts 5× (Sigma-Aldrich), 10 mL glucose 20% (w/v), 1 mL MgSO4.7H20 (1 M), 1 mL thiamine hydrochloride (1 mg/mL), 1 mL $CaCl_2$ (1 M) in 1 L purified water. Media was supplemented with 10% v/v LB (lysogeny broth). The pH was 7.0 at 25 °C.

## Bacterial culture

*E. coli* DH5α cell cultures (150 mL) were grown in M9LB bacterial growth media at 37 °C under low light conditions and constant shaking (250 RPM).

## Growth curve assay

Overnight cultures of *E. coli* DH5α were washed once in phosphate-buffered saline solution (PBS; Thermo Scientific), and pelleted. The supernatant was removed, and cells resuspended culture media and inoculated in 150 mL to an initial optical density ($OD_{600}$) of 0.05. Cell cultures were harvested at OD 0.1, 0.4 and 1.4 to sample for lag, exponential and stationary phase, respectively.

## Knockout growth curve phenotyping

*E. coli* BW25113 gene knockouts of interest were isolated from the KEIO collection (Baba et al, 2006) with the respective genes disrupted by the replacement of the ORF by the KanR cassette. Overnight cultures of the *E. coli* gene knockouts were grown on LB and then washed with PBS to remove residual LB. The washed cells were then used to inoculate into 96-well plates to an $OD_{600}$ of 0.05 in either M9LB or M9 glycerol (48 mM $Na_2HPO_4$, 22 mM $KH_2PO_4$, 19 mM $NH_4Cl$, 9 mM NaCl, 2 mM $MgSO_4$, 1% glycerol). Triplicate cultures were grown at 37 °C, and shaken in an Epoch2 microplate spectrophotometer (BioTek). $OD_{600}$ measurements were taken every hour up to 50 h.

Imaging was carried out on $2 \times 2$ mm agar pads produced by sandwiching a solution of 1.5% molten agarose in medium (LB or Gly) between two microscope slides. As a mould for the pads and to ensure that their thickness was uniform, two gene frames (Thermo FIsher, AB0576) were stacked between the microscope slides. When solid, the 1x1cm pads were split manually with a scalpel and

positioned in an imaging chamber made up of two coverslips held together by a larger gene frame (Thermo Fisher, AB0577). Cells were then transferred to a Nikon Eclipse Ti-E inverted microscope preheated at 37 °C and automatically imaged in phase contrast for up to 8 h using a Genicam (Teledyne FLIR BFS-U3-70S7M-C) and a ×60 magnification (oil objective, numerical aperture 1.45) for a 0.1067-pixel size. Phase contrast images were segmented using SuperSegger (Stylianidou et al, 2016).

## Bacterial N-terminal epitope tagging

Proteins of interest were amplified by PCR from *E. coli* DH5α and inserted into the expression vector pBADcLIC (Geertsma et al, 2008) by recombination-based cloning, which introduces a C-terminal tag (-ENLYFQGHHHHHHHHHHH). The resulting plasmids were then transformed into *E. coli* DH5α for downstream studies.

## Orthogonal organic phase separation (OOPS)

Cell cultures were grown in 150 mL of M9LB media overnight (~16 h), or to the indicated optical density for dynamic experiments. In all, 10exp7 cells were used per replica and condition. OOPS was performed according to (Villanueva et al, 2020); briefly: In non-cross-linked (NC) controls, cells were immediately pelleted at $6000 \times g$, and the supernatant was removed and placed on ice. In cross-linked (CL) samples, UV cross-linking was performed directly on the culture by UV irradiation at 254 nm with 700 mJ/cm$^2$ (CL-1000 Ultraviolet Crosslinker; UVP). Immediately after cross-linking cells were pelleted and the supernatant removed. Cell pellets (both NC and CL) were resuspended in 50 µL acidic guanidinium-thiocyanate-phenol (Trizol, Thermo Fisher Scientific), and 500 µL of 0.5-mm glass beads (Sigma-Aldrich) were added to each Eppendorf. Cell lysates were disrupted by two rounds of lysis at 6 m/s for 60 s (MP FastPrep-24 5 G Tissue Homogenizer; MP Biomedicals). In between runs, samples were placed on ice to avoid warming. A further 1 mL of Trizol was added to each sample and placed on a table-top shaker for 5 min at 1100 RPM. To clear the lysate, samples were centrifuged for 5 min at $6000 \times g$ at 4 °C and the supernatant was transferred to a new tube. If total proteome was assayed, an aliquot of 100 µL was taken from the homogenised lysate for TMT labelling. To recover the RNA-binding proteins (RBPs), 200 µL of chloroform (Fisher Scientific) was added to each sample, phases were vortexed, and the sample was centrifuged for 15 min at $12,000 \times g$ at 4 °C. The upper, aqueous phase (containing non-cross-linked RNAs) and the lower, organic phase (containing non-cross-linked proteins) were discarded. Interphase (containing the protein–RNA adducts) was subjected to two additional rounds of Trizol phase separation, precipitated by the addition of nine volumes of methanol, and pelleted by centrifugation at $14,000 \times g$ RT for 10 min with the exception of samples for the RNase control assay.

## RNase treatment of OOPS interphases

For RNase control samples, three rounds of phase separation were performed, and the resulting interphase was resuspended in 1 mL of RNase-free water. Suspension was slowly mixed by inversion and centrifuged at max speed for 2 min. The supernatant (RNase-free water and any remaining organic phase) was discarded, and the pellet was resuspended in 500 µL of 1% SDS in RNase-free water. The interphase was solubilised by gentle pipetting and 100 µL 3 M sodium acetate (Invitrogen) was added. Samples were then vortexed and 600 µL isopropanol added, placed on ice for 10 min and centrifuged at max speed for 10 min. Once again, the supernatant was discarded, and the pellet was washed first with 100% ethanol and then with 70% ethanol. Supernatant newly removed and pellets resuspended in 200 µL RNase-free water. 90 µL of the solubilised interface was transferred to a new tube and 10 L 10× RNase control buffer (100 mM Tris-HCl pH 7.5, 3 M NaCl, 50 mM EDTA, 10× DTT) was added; this is the RNase-untreated sample. Another 90 µL of solubilised interface was transferred to a new tube and 10 µL 10X RNase digestion buffer (100 mM Tris-HCl pH 7.5, 3 M NaCl, 50 mM EDTA) added; this is the RNAse-treated sample. All samples were incubated at 60 °C for 10 min to inactivate any potential RNase contamination, and 2 µL of RNase A/T1 mix (Thermo Scientific, EN0551) was added to the RNAse-treated samples. All samples were incubated for 4 h at 400 RPM and at 37 °C. One millilitre of Trizol was added and samples were homogenised by vortexing. As per standard OOPS workflow, 200 µL of chloroform (Fisher Scientific) were added, phases were vortexed, and the sample was centrifuged for 15 min at $12,000 \times g$ at 4 C. Interface (containing the protein–RNA adducts) was subjected to extra Trizol phase separation cycles, precipitated by addition of nine volumes of methanol, and pelleted by centrifugation at $14,000 \times g$ for 10 min at RT.

## Proteomic sample preparation

Pelleted interphases were resuspended in 100 µL of 100 mM TEAB (Sigma-Aldrich) and reduced with 20 mM DTT (Sigma-Aldrich) for 60 min at RT and alkylated with 40 mM iodoacetamide (IAA, Sigma-Aldrich) in the dark for at least 60 min at RT. Samples were digested with 1 µg trypsin (Promega) overnight at 37 °C, with the exception of samples for TMT labelling, which were digested with 1 µg Lys-C (Promega) overnight at 37 °C. Subsequently, 1 µg of modified trypsin (Promega) was added, and the samples were incubated for 3–4 h at 37 °C. Samples were then acidified with trifluoroacetic acid (TFA, 0.1% (v/v) final concentration; Sigma-Aldrich) and centrifuged for 10 min at $21,000 \times g$ and RT. The supernatant was frozen at −80 °C until required.

TMT samples: For peptide clean up and quantification, 200 µL of Poros Oligo R3 (Thermo Fisher Scientific) resin slurry (~150 µL resin) was packed into Pierce centrifuge columns (Thermo Fisher Scientific) and equilibrated with 0.1% TFA. Samples were loaded, washed three times with 150 µL 0.1% TFA and eluted with 300 µL 70% acetonitrile (ACN). 10 µL from each elution was removed for Qubit protein quantification assay (Thermo Fisher Scientific) performed according to the manufacturer's protocol, with the remaining sample retained for MS.

## LC-MS/MS

For label-free quantification (LFQ), injected samples were analysed using an Ultimate 3000 RSLC$^{TM}$ nanosystem (Thermo Scientific, Hemel Hempstead) coupled to an Orbitrap Eclipse$^{TM}$ mass spectrometer (Thermo Scientific). The sample was loaded onto the trapping column (Thermo Scientific, PepMap100, C18,

300 μm × 5 mm), using partial loop injection for 3 min at a flow rate of 15 μL/min with 0.1% (v/v) formic acid in 3% acetonitrile. The sample was resolved on the analytical column (uPAC 200 cm column) at a flow rate of 300 nL/min using a gradient of 97% A (0.1% formic acid) 3% B (80% acetonitrile 0.1% formic acid) to 35% B over 340 min, then to 90% B for an additional 5 min, then to 90% B for another 4 min, percentage of B was then lowered to 3% to allow the column to re-equilibrate for 20 min before the next injection. The data-dependent programme used for data acquisition consisted of a 120,000-resolution full-scan MS scan (AGC set to 100% (4E6 ions) with a maximum fill time of 50 ms). MS/MS was performed at the mass range of 150–2000 $m/z$ using a resolution of 60,000 (AGC set to 100% (1E5 ions) with a maximum fill time of 118 ms) with an isolation window of 1.2 $m/z$ and an HCD collision energy of 30%.

TMT-10plex (Thermo Fisher Scientific) labelling from desalted peptides was performed according to the manufacturer's protocol. Equal amounts of desalted peptides were labelled immediately after being quantified with Qubit protein assay (Thermo Fisher Scientific) as per the manufacturer's instructions. Multiplexed TMT samples were separated into four fractions using the Pierce High pH Reversed-Phase Peptide Fractionation Kit (Thermo Fisher Scientific). TMT labelled fractions were analysed in an Orbitrap Fusion Lumos using a resolution of 50,000. Mass spectra were acquired in positive ion mode applying data acquisition using synchronous precursor selection MS3 (SPS-MS[3]) acquisition mode (McAlister et al, 2014). Data deposited in PRIDE (Vizcaíno et al, 2016) (accession number: PXD043373).

## MS spectra processing and peptide and protein identification

Raw data were viewed in Xcalibur v2.1 (Thermo Fisher Scientific), and data processing was performed using Proteome Discoverer v2.1 (Thermo Fisher Scientific). The raw files were submitted to a database search using Proteome Discoverer with Mascot, SequestHF and MS Amanda (Dorfer et al, 2014) algorithms against the *E. coli* K-12 DH5α database downloaded in early 2017, containing *E. coli* protein sequences from UniProt/Swiss-Prot and UniProt/TrEMBL. Common contaminant proteins (e.g., porcine trypsin) identified were removed from the results lists before further analysis, except in the case of the RNAse assay, where porcine trypsin was kept for normalisation. The spectra identification was performed with the following parameters: MS accuracy, 10 p.p.m.; MS/MS accuracy of 0.05 Da for spectra acquired in Orbitrap analyser and 0.5 Da for spectra acquired in Ion Trap analyser; up to two missed cleavage sites allowed; carbamido-methylation of cysteine (as well as TMT-10plex tagging of lysine and peptide N terminus for TMT labelled samples) as a fixed modification; and oxidation of methionine and deamidated asparagine and glutamine as variable modifications. Percolator node was used for false discovery rate estimation and only rank 1 peptide identifications of high confidence (FDR < 1%) were accepted. A minimum of two high-confidence peptides per protein was required for identification using Proteome Discoverer.

TMT reporter values were assessed through Proteome Discovery v2.1 using the Most Confident Centroid method for peak integration and integration tolerance of 20 p.p.m. Reporter ion intensities were adjusted to correct for the isotopic impurities of the different TMT reagents (manufacturer's specifications) (Dataset EV8).

## Proteomics bioinformatics and data analysis

Peptide-level output from Proteome Discoverer was reprocessed with the 'add_master_protein.py' script (https://github.com/TomSmithCGAT/CamProt) to ensure uniform peptide-to-protein assignments for all samples from a single experiment and identify peptides that are likely to originate from contaminating proteins such as keratin. For quantitative experiments, peptide-level quantification was obtained by summing the quantification values for all peptides with the same sequence but different modifications. Protein-level quantification was then obtained by applying the 'robust' approach from the MSnbase R package (Gatto and Lilley, 2012). For the growth curve experiments ($n = 5$ for each phase), the protein abundance was quantified by label-free quantification (LFQ) and data analysis was performed using the pRoloc R package (Gatto et al, 2014). For TMT experiments, data analysis was performed using the MSnbase R package (Gatto et al, 2014). Log2-transformed protein abundance was centre-median normal-ised within each sample. For the RNase assay, the ratio between treated and untreated protein abundance was calculated for each sample separately and aggregated to average protein ratio. Samples were normalised against the input porcine trypsin (UNIPROTKB ID: P00761) to correct for any downstream pipetting error.

## iCLIP

iCLIP protocol was applied as described in Lee et al, 2021. Briefly, bacterial strains transformed with tagged protein of interest (see 'Bacterial N-terminal epitope tagging'), as well as a wild-type (WT) strain, were induced with 0.01% L-arabinose at the exponential phase (OD ~0.4) and harvested at stationary phase and irradiated with UV-C light for the cross-linked samples (CL) or directly pelleted for the non-cross-linked (NC). Here, the WT strain was the negative control and was UV-irradiated. Cell suspensions were cleared once, pelleted and resuspended in iCLIP lysis buffer (50 mM Tris-HCl pH 7.4, 100 mM NaCl, 1% Igepal CA-630 (Sigma I8896), 0.1% SDS, 0.5% sodium deoxycholate, 1× cOmplete protease inhibitor cocktail, RNasin) and cells were lysed as detailed above. To trim the cross-linked RNA, 2 μL TURBO DNase (AM2238 Thermo Fisher Scientific) and 1 μL diluted RNAse (1:100–1:6000 dilution from 10 U/μL stock EN0602, Thermo Fisher Scientific) were added to the cellular extracts. The solution was incubated at 37 °C for 3 min and immediately transferred on ice. Target proteins of interest were purified by incubating lysates overnight with His-tag antibody (CST #2366) conjugated to Protein G Dynabeads. Dephosphorylation of cyclic phosphate groups were carried out with T4 PNK (10 U/μL, M0201, NEB) in a low pH buffer (5× PNK pH 6.5 buffer, PNK,FastAP alkaline phosphatase (EF0654, Thermo Fisher Scientific), RNAsin) for 40 min at 37 °C. Pre-adenylated L3-1R-App 3' adaptors were ligated using T4 RNA ligase 1 (M0204, NEB) for 75 min at 25 °C. Excess adaptors were removed by RECJ (NEB M0264S) and 5'-deadenylase (NEB M0331S) treatments. RNA–protein complexes were isolated by SDS-PAGE and nitrocellulose transfer. RNA was then released from the membrane by Proteinase K treatment and purified by

phenol-chloroform extraction. Isolated RNA was reverse transcribed with SuperScript IV (Invitrogen, 18090010) and after reverse transcription, RNA was degraded by alkaline hydrolysis with NaOH, underwent a bead-based purification and circularised with CircLigase (Epicentre/Illumina, CL9021K) for 2 h. Circularised cDNA was directly PCR amplified, quantified with Bioanalyzer and sequenced on Illumina MiSeq. Adaptor and primer sequences listed in Dataset EV9. Data deposited in GEO (accession number: GSE235661).

## iCLIP analysis

iCLIP data were demultiplexed using iCount (version 2.0.0) de-multiplex function (https://github.com/tomazc/iCount) according to the sample barcodes (Dataset EV9). Demultiplexed files were uploaded to https://app.flow.bio/ and run on the CLIP-seq v1.1 pipeline (https://github.com/goodwright/clipseq/tree/1.1) using the genome found at https://www.ncbi.nlm.nih.gov/datasets/genome/GCF_000005845.2/ and pre-mapped against an artificial genome containing all tRNAs found at http://gtrnadb.ucsc.edu/genomes/bacteria/Esch_coli_K_12_MG1655/eschColi_K_12_MG1655-mature-tRNAs.fa Execution was run with default parameters except in the case of STAR, where the following arguments were detailed: *--outFilterMultimapNmax 100 --outFilterMultimapScoreRange 1 --outSAMattributes All --alignIntronMin 1000000 --outFilterScoreMin 10 --alignEndsType Extend5pOfRead1*. BAMs were filtered for uniquely mapped reads and cross-link sites were taken as read start - 1 position. Peaks were called using iCount-Mini v2.0.3 (https://github.com/ulelab/icount-mini). The resulting '*.peaks.bed* files were downloaded for downstream analysis. All peaks identified in the negative control were blacklisted using bedtools (version 2.31.1) *intersect -v*. The remaining peaks were intersected among the CL replicates using bedtools command *intersect -wo -f 0.1 -r* and *intersect -u*. For each peak the median score and peak size was computed across the replicates. Comparative visualisation of iCLIP data was generated with the CLIPplotR package (Chakrabarti et al, 2023), which takes cross-linked sites detected as input, and normalises them by library size and plots over regions of interest. For the detection of binding motifs, Positionally enriched k-mer analysis (PEKA) (Kuret et al, 2022) was completed in the CLIP-seq v1.1 pipeline followed by STREME analysis on the peak sequences with the parameters *--verbosity 1 --oc. --dna --totallength 4000000 --time 14400 --minw 8 --maxw 15 --thresh 0.05 --align center*, where the NC peaks were used as the negative control.

## Statistics

Data handling was performed with R v4.0.2 using tidyverse (Wickham et al, 2019). Plotting was performed with the ggplot2 R package (Wickham, 2009).

For the RNase assay experiments, only proteins present in both treated and untreated conditions were considered. For the rest of the proteins, all observations across the replicates were treated as independent observations. DeqMS R package (Zhu et al, 2020) was applied to model the abundance as a function of the condition (RNase-treated or untreated), where log2 median RNase/Control ratio >0 (enriched) or <0 (depleted), with a BH-adjusted $P$ value < 0.01 considered significant. Proteins with fewer than 2 hits per condition were excluded from the statistical test because of insufficient power.

For the dynamic growth curve experiment by label-free quantification (LFQ), proteins with a change of abundance were identified using the 'lm' function in R (Ritchie et al, 2015). Specifically, to identify protein with a change in abundance between two growth phases, total abundance was modelled as a function of the growth phase alone (abundance ~ growth phase). The $P$ values for the growth curve coefficients for each protein were adjusted to account for multiple hypothesis testing according to Benjamini and Hochberg (Benjamini and Hochberg, 1995) and proteins with an adjusted $P < 0.01$ (1% FDR) were considered significant. For the heatmap representation, protein abundances were $z$-score normalised within the total samples. Hierarchical clustering was performed with the R hclust function using Pearson correlation as the distance metric and average linkage.

For the dynamic growth curve experiment by TMT quantification, proteins with a change of abundance or RNA binding were identified using the lm function in R. Specifically, to identify protein with a change in abundance between two growth phases, total abundance was modelled as a function of the growth phase alone (abundance ~ growth*phase). The $P$ values for the growth curve coefficients for each protein were adjusted to account for multiple hypothesis testing according to Benjamini and Hochberg (Benjamini and Hochberg, 1995) and proteins with an adjusted $P < 0.01$ (1% FDR) were considered to have changed abundance. To identify proteins with a change in RNA binding between two growth phases, protein abundance in the total proteome and OOPS samples was modelled as a function of the growth phase, the extraction type ('total' or 'OOPS') and the interaction between these two variables (abundance growth phase + type + [growth phase * type]). The interaction term denotes whether: (i) The abundance in 'total' and 'OOPS' follow the same pattern across the growth phases (coefficient is zero), indicating total abundance determined the amount of protein bound to RNA; or (ii) The abundance in 'total' and 'OOPS' diverge (nonzero coefficient), indicating a change in RNA binding between the phases. The $P$ values for the interaction term were obtained and adjusted as indicated above. For the heatmap representation, protein abundances were z-score normalised within the total and OOPS samples separately. Hierarchical clustering was performed with the R hclust method using Pearson correlation as the distance metric.

## Physicochemical property analysis of RBPs

Protein properties were obtained with the 'Peptides' R package (version 2.4.5) (Osorio et al, 2015). Normality between the groups was assumed due to the sample size (>60 data points per group). Variance was assessed visually as well as formally by the 'leveneTest' function from the car R and a $t$ test was performed to assess significance.

## Protein interaction network analysis

Proteins queried in the StringApp on the Cytoscape v3.9.1 interface (Doncheva et al, 2019, 2023). For physical subnetworks, proteins queried against the *Escherichia coli* K-12 substr. MG1655 pre-loaded database and filtered above 0.4 in confidence score value.

## Ortholog analysis

The hierarchical orthologous groups (HOGs) for each candidate protein were retrieved from the OMA browser (Altenhoff et al, 2021) (Dataset EV10), these are sets of genes that are inferred to have descended from a common ancestral gene. Resulting sets were intersected at the genus level with the interactive Tree of Life (iTOL) version 5 (Letunic and Bork, 2021) containing 191 unique tips across the three Domains of life, using R custom scripts. Code available upon request.

## Structural conservation analysis

Uniprot IDs of the candidate proteins were used to retrieve all their associated PDB IDs using the Uniprot Retrieve/ID mapping tool. If no experimentally resolved structures were available, AlphaFold predictions were used (Jumper et al, 2021; Varadi et al, 2022). Orthologs were defined via the InParanoid database (Sonnhammer and Östlund, 2015) and structures aligned with the following command: 'align bacterial_protein human_protein' on PyMOL Version 2.5.2, where possible experimentally resolved structures were used. Root Mean Square Deviation (RMSD) calculated as defined in script at https://pymolwiki.org/index.php/ColorByRMSD and the distances between aligned C-alpha atom pairs are stored as B-factors of these residues, command run: 'colorbyrmsd human_protein, bacterial_protein, doAlign=1, doPretty=1'. Structures then coloured by RMSD using 'spectrum b, blue_white_red, minimum=0, maximum=10', with a ramp included with 'ramp_new ramp, POI, [0, 5, 10], [blue, white, red]' where POI stands for 'protein of interest'. Highlighted in the structure are the predicted RNA-binding motifs by RBDpep (Castello et al, 2016) and/or OOPS direct evidence (Queiroz et al, 2019), found by aligning protein sequences with online tool Clustal Omega (https://www.ebi.ac.uk/Tools/msa/clustalo/) (Sievers et al, 2011).

# Data availability

The datasets produced in this study are available in the following datasets: (1) Mass spectrometry proteomics data: deposited to the ProteomeXchange Consortium via the PRIDE (Vizcaíno et al, 2016) partner repository with the dataset identifier PXD043373. (2) iCLIP sequencing data: deposited to the Gene Expression Omnibus (GEO), accession code GSE235661.

# Peer review information

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

## Acknowledgements

The authors would like to thank Dr Caia DS Duncan and Prof Juan Mata for kindly letting us access the MP FastPrep-24 5 G Tissue Homogenizer (MP Biomedicals) and Prof Pietro Cicuta for microscopy access. Mass spectrometry analysis was performed at the Proteomics Facility of the Medical Research Council Toxicology Unit University of Cambridge, Cambridge, UK by Catarina Franco and Rayner Queiroz. The authors additionally thank Mike Deery at CCP for his assistance with MS sample preparation and analysis. The authors also thank Dr. Maria Marti Solano for her assistance during the manuscript writing and Dr. Xiaoteng Jiang for his inputs and manuscript proofreading. In addition, Bini Ramchandran, Yagnesh Umrania and Julie Howard Murkin for help in proteomic file management and submission. Finally, the authors thank Dr. Sergey Moshkovksiy and Prof. Urlaub Henning for their help and support in interpreting the MS data. MM is supported by the Medical Research Council, grant number 5TR00. LM is supported by a Herchel Smith Postdoctoral. CSD is supported by a Herchel Smith PhD Research Studentship. KD is supported by UK DRI at King's College London. EV was supported by Wellcome Trust, grant numbers 110071/Z/15/Z awarded to KSL. The contribution of KD, CC and JU research was funded by the European Union's Horizon 2020 research and innovation programme (835300-RNPdynamics). The Francis Crick Institute receives its core funding from Cancer Research UK (FC001110), the UK Medical Research Council (FC001110), and the Wellcome Trust (FC001110). RH and GHT were supported by InnovateUK and the Biotechnology and Biological Sciences Research Council through the Industrial Biotechnology Catalyst grant BB/N01040X/1. For the purpose of Open Access, the authors have applied a CC BY public copyright licence to any Author Accepted Manuscript version arising from this submission.

## Author contributions

**Mie Monti**: Conceptualization; Resources; Data curation; Software; Formal analysis; Validation; Investigation; Visualization; Methodology; Writing—original draft; Writing—review and editing. **Reyme Herman**: Validation; Investigation; Visualization. **Leonardo Mancini**: Conceptualization; Validation; Investigation; Visualization; Methodology; Writing—review and editing. **Charlotte Capitanchik**: Data curation; Formal analysis; Investigation; Methodology; Writing—review and editing. **Karen Davey**: Investigation; Methodology. **Charlotte S Dawson**: Supervision; Funding acquisition; Investigation; Writing—original draft; Writing—review and editing. **Jernej Ule**: Funding acquisition; Writing—original draft; Writing—review and editing. **Gavin H Thomas**: Supervision; Funding acquisition; Writing—original draft; Writing—review and editing. **Anne E Willis**: Conceptualization; Supervision; Funding acquisition; Investigation; Visualization; Methodology; Writing—original draft; Project administration; Writing—review and editing. **Kathryn S Lilley**: Conceptualization; Funding acquisition; Validation; Investigation; Visualization; Writing—original draft; Writing—review and editing. **Eneko Villanueva**: Conceptualization; Data curation; Formal analysis; Supervision; Investigation; Visualization; Methodology; Writing—original draft; Project administration; Writing—review and editing.

## Disclosure and competing interests statement

The authors declare no competing interests. JU is an editorial advisory board member. This has no bearing on the editorial consideration of this article for publication.

