## [Peer Review File · Molecular Systems Biology]

Interrogation of RNA-protein interaction dynamics in bacterial growth

Mie Monti, Reyme Herman, Leonardo Mancini, Charlotte Capitanchik, Karen Davey, Charlotte Dawson, Jernej Ule, Gavin Thomas, Anne Willis, Kathryn Lilley, and Eneko Villanueva

Corresponding author(s): Eneko Villanueva (ev318@cam.ac.uk) , Kathryn Lilley (k.s.lilley@bioc.cam.ac.uk), Anne Willis (aew80@mrc-tox.cam.ac.uk)

Review Timeline:

Submission Date:	2nd Oct 23
Editorial Decision:	3rd Nov 23
Revision Received:	26th Jan 24
Editorial Decision:	4th Mar 24
Revision Received:	8th Mar 24
Accepted:	11th Mar 24

Editor: Maria Polychronidou

Transaction Report:

3rd Nov 2023

Manuscript Number: MSB-2023-12032

Title: Interrogation of RNA-protein interaction dynamics in bacterial growth

Dear Eneko,

Thank you again for submitting your work to Molecular Systems Biology. We have now heard back from the three reviewers who agreed to evaluate your study. The reviewers are positive about the goals of the study. However, as you will see below, they raise a series of concerns, which we would ask you to address in a revision. Reviewer #2 is concerned about the lack of depth in terms of follow up analyses of the identified RNA-protein interactions. While we acknowledge this concern, given the potential resource value of the study for the community, as indicated by the other two referees, we think that this concern does not preclude publication in MSB.

The comments of the referees are rather clear and seem straightforward to address so I think there is no need to repeat any of them here. All issues raised by the referees would need to be satisfactorily addressed. Please let me know in case you would like to discuss in further detail any of the issues raised, I would be happy to schedule a call.

On a more editorial level, we would ask you to address the following points:

- Please provide a .doc version of the manuscript text (including legends for the main figures) and individual production quality figure files for the main Figures (one file per figure).
- Please include 5 keywords.
- We have replaced Supplementary Information by the Expanded View (EV format). In this case, all additional figures can be included in a PDF called Appendix. Appendix figures should be labeled and called out as: "Appendix Figure S1, Appendix Figure S2... Appendix Table S1..." etc. Each legend should be below the corresponding Figure/Table in the Appendix. Please include a Table of Contents in the beginning of the Appendix. For detailed instructions regarding expanded view please refer to our Author Guidelines: .
- Tables S1-S7 should be provided and called out in the text as Datasets EV1-EV7. Please provide one file per EV Dataset. Please include the description of each EV Dataset in the dataset file itself, i.e. in a separate tab for .xls files or as a README.txt file in .zip folders for .csv files.
- Please provide a "standfirst text" summarizing the study in one or two sentences (approximately 250 characters), three to four "bullet points" highlighting the main findings and a "synopsis image" (550px width and max 400px height, jpeg format) to highlight the paper on our homepage.
- Please include a "Disclosure and Competing Interests statement" in the main text.
- All Materials and Methods need to be described in the main text. We would encourage you to use 'Structured Methods', our new Materials and Methods format. According to this format, the Material and Methods section should include a Reagents and Tools Table (listing key reagents, experimental models, software and relevant equipment and including their sources and relevant identifiers) followed by a Methods and Protocols section in which we encourage the authors to describe their methods using a step-by-step protocol format with bullet points, to facilitate the adoption of the methodologies across labs. More information on how to adhere to this format as well as downloadable templates (.doc or .xls) for the Reagents and Tools Table can be found in our author guidelines: . An example of a Method paper with Structured Methods can be found here:
- Please include a Data availability section describing how the data, code etc. have been made available. This section needs to be formatted according to the example below:
The datasets and computer code produced in this study are available in the following databases:
 - Chip-Seq data: Gene Expression Omnibus GSE46748 (<https://www.ncbi.nlm.nih.gov/geo/query/acc.cgi?acc=GSE46748>)
 - Modeling computer scripts: GitHub (<https://github.com/SysBioChalmers/GECKO/releases/tag/v1.0>)
 - [data type]: [full name of the resource] [accession number/identifier] ([doi or URL or identifiers.org/DATABASE:ACCESSION])
- For data quantification: please specify the name of the statistical test used to generate error bars and P values, the number (n) of independent experiments (specify technical or biological replicates) underlying each data point and the test used to calculate p-values in each figure legend. The figure legends should contain a basic description of n, P and the test applied. Graphs must include a description of the bars and the error bars (s.d., s.e.m.).
- The References should be formatted according to the Molecular Systems Biology reference style (i.e., ordered alphabetically

and listing the first 10 authors followed by et al).

- When you resubmit your manuscript, please download our CHECKLIST (<https://bit.ly/EMBOPressAuthorChecklist>) and include the completed form in your submission.

Please note that the Author Checklist will be published alongside the paper as part of the transparent process (<https://www.embopress.org/page/journal/17444292/authorguide#transparentprocess>).

If you feel you can satisfactorily deal with these points and those listed by the referees, you may wish to submit a revised version of your manuscript. Please attach a covering letter giving details of the way in which you have handled each of the points raised by the referees. A revised manuscript will be once again subject to review and you probably understand that we can give you no guarantee at this stage that the eventual outcome will be favorable.

Kind regards,

Maria

Maria Polychronidou, PhD
Senior Editor
Molecular Systems Biology

We realize that it is difficult to revise to a specific deadline. In the interest of protecting the conceptual advance provided by the work, we recommend a revision within 3 months (1st Feb 2024). Please discuss the revision progress ahead of this time with the editor if you require more time to complete the revisions. Use the link below to submit your revision:

IMPORTANT: When you send your revision, we will require the following items:

1. the manuscript text in LaTeX, RTF or MS Word format
2. a letter with a detailed description of the changes made in response to the referees. Please specify clearly the exact places in the text (pages and paragraphs) where each change has been made in response to each specific comment given
3. three to four 'bullet points' highlighting the main findings of your study
4. a short 'blurb' text summarizing in two sentences the study (max. 250 characters)
5. a 'thumbnail image' (550px width and max 400px height, Illustrator, PowerPoint or jpeg format), which can be used as 'visual title' for the synopsis section of your paper.
6. Please include an author contributions statement after the Acknowledgements section (see <https://www.embopress.org/page/journal/17444292/authorguide>)
7. Please complete the CHECKLIST available at (<https://bit.ly/EMBOPressAuthorChecklist>).

Please note that the Author Checklist will be published alongside the paper as part of the transparent process (<https://www.embopress.org/page/journal/17444292/authorguide#transparentprocess>).

See also figure legend guidelines: <https://www.embopress.org/page/journal/17444292/authorguide#figureformat>

9. Please note that corresponding authors are required to supply an ORCID ID for their name upon submission of a revised manuscript (EMBO Press signed a joint statement to encourage ORCID adoption).

(<https://www.embopress.org/page/journal/17444292/authorguide#editorialprocess>)

Currently, our records indicate that the ORCID for your account is 0000-0002-3585-8846.

Link Not Available

The system will prompt you to fill in your funding and payment information. This will allow Wiley to send you a quote for the article processing charge (APC) in case of acceptance. This quote takes into account any reduction or fee waivers that you may be eligible for. Authors do not need to pay any fees before their manuscript is accepted and transferred to the publisher.

EMBO Press participates in many Publish and Read agreements that allow authors to publish Open Access with reduced/no publication charges. Check your eligibility: <https://authorservices.wiley.com/author-resources/Journal-Authors/open-access/affiliation-policies-payments/index.html>

*** PLEASE NOTE *** As part of the EMBO Press transparent editorial process initiative (see our Editorial at <https://dx.doi.org/10.1038/msb.2010.72>), Molecular Systems Biology publishes online a Review Process File with each accepted manuscripts. This file will be published in conjunction with your paper and will include the anonymous referee reports, your point-by-point response and all pertinent correspondence relating to the manuscript. If you do NOT want this File to be published, please inform the editorial office at msb@embo.org within 14 days upon receipt of the present letter.

Reviewer #1:

Monti et al. apply here OOPS to study the RBPome of bacteria and the changes that it undergoes during the different cell growth phases. OOPS is very relevant to prokaryotes, because in contrast to eukaryotic mRNAs, bacteria are not polyadenylated and are, therefore, not readily accessible to oligo(dT)-based approaches. It is nice though to see that the presented data overlap very well with previous RBPomes in bacteria while providing substantial newly identified RBPs. Moreover, they study the dynamics of these RBPs during the different growth phases showing that the bacterial RBPome is highly dynamic, as previously shown in mammalian cells. The authors define a group of novel RBPs that are differentially regulated during bacterial life and provide evidence of binding to several RNAs.

The author team is composed by experts in the field, and as such the paper is well written and the data seems solid. There are though several points that can be improved and will strengthen this work.

Major points:

- As a general comment, I think adding page numbers and lane numbers would help a lot in the revision of the manuscript.
- The introduction does not read as an introduction and indeed the first paragraph is the only one that provide context to the work. The rest is a repetition of the experiments done in this work. I suggest that this must be better balanced expanding the knowns on the field either from the RNA metabolism in bacteria angle or by explaining how other studies have profiled the dynamics of the eukaryotic (but not prokaryotic) interactome. Either angle would add value to the work by contextualising it better.
- Authors do not explore one of the main limitations and source of bias of their approach, which is the composition of the transcriptome they purify from bacteria. In order to understand the sources that contribute to the RBPome, it is important to define what the relative abundance of each RNA biotype is in the pull down. An RNAseq analysis of the eluates considering multiple mapping genes (e.g. rRNA) would help to understand the composition of the isolated transcriptome in bacteria and the relative contribution of each RNA biotype to the final results.
- The number of replicates in the different experiments appears to be inconsistent or potentially mislabelled. In Fig 1c it is indicated that there were 3 biological replicates, but the figure shows 5 columns in each growth stage, suggesting 5 replicates rather than 3. Maybe I have interpreted something wrongly, but I think this needs further clarification. In addition, Fig 5.f-g shows 1 control replicate per condition and 2 biological replicates for the samples. If that is the experimental design and is not misinterpreted by me, then the number of replicates in that experiment is not statistically robust enough.
- Bona fide RBPs have physicochemical properties that make them suitable for RNA binding. This is particularly easy to test in bacteria, as proteins are in average smaller than their eukaryotic counterparts and are more globular. Authors should examine general properties such as hydrophobicity (in RBPs because the accumulation of basic residues) and enrichment for amino acids such as K, R, Y, and others frequently found at protein-RNA interfaces. There have been many studies looking at these amino acid biases in detail. Authors can analyse these parameters for the total proteome, the total RBPome, the previously known RBPs and the newly identified RBP group. Finding similar patterns would suggest that these groups are all similar biochemically and thus compatible with RNA binding. This would increase the value of the dataset by benchmarking it to known features of RBPs. If alphafold or crystal structures exist, it could be possible to use software such as BindUP that predict RNA-binding surfaces because their physicochemical properties. Such approach could be applied to their candidates (discussed in the last part of the paper) to test if there are a RNA-binding compatible surface and if so, where.
- Authors identified 3 RBPs with conserved orthologs between *E.coli* and *H.sapiens* including YhgF. It is mentioned the lack of characterisation of YhgF; however, a recent paper showed through its binding profile (CLIP-seq) its potential involvement in transcriptional regulation. I feel expanding the text and cross-referencing to the available CLIP would solidify their results (if in agreement). Also, do the orthologs of these proteins in human bind to RNA? Do they have CLIP-seq datasets? How do they compare to bacterial data? What do we know about them that can be used to understand better the bacterial counterpart?
- Authors use a STRING-db prediction for YfiF and HtpG interaction network. Are there studies determining it experimentally? What is it known of the partners of the cellular orthologs?
- Further characterization of the CLIPs of newly discovered proteins would improve the value of this work. The authors show binding sites on the overall cellular transcriptome and maps of a few selected transcripts. I feel this CLIP analysis is treated very superficially. Which binding sites are statistically enriched? Do these proteins recognize any specific sequence? Do they bind any signature within the target RNAs? For example, for the tRNAs, do they recognize a particular stem-loop? Plots showing binding site density across the different nucleotides of the tRNAs could help to define the binding signature of the protein. It

- seems to me that the CLIP-seq work is underexplored and after putting so much effort into it, I feel should be analysed deeper.
- Following on the CLIP-seq experiments, the authors used MiSeq sequencers for their CLIP sequencing that produce only a few million reads per lane. Hence, the read depth provided by a MiSeq does not seem sufficient for a comprehensive CLIP analysis. In other bacterial CLIP experiments, Illumina NextSeq sequencers have been used. Could the authors specify how many samples were run per run and what read depth they accomplished? Are these sufficient for robust statistical analyses?
 - Authors showed that ablation of their candidate proteins impair cellular growth, however, it isn't clear if this is due to intrinsic toxicity or impaired cell division. Also, no connection with their RNA binding activities is provided. I think this part can be expanded with a few selected experiments (e.g. do the bacteria look normal under the microscope? Can they divide normally?) or supporting their data with preexisting research (if available).
 - Fig 5.f-g. The authors do not describe or mention either of these figures in the text.

MINOR POINTS:

- Abstract: 'provide the first dynamic RBPome'; I don't think this is correct > 'for the first time the dynamics of the bacterial RBPome'
- 'This unbiased method' > I would recommend authors to avoid this assumption as the method is biased. First, UV crosslinking would favour proteins that bind to single stranded regions, second, mRNAs are a small subpopulation in the context of total RNA. Most methods are biased, and, therefore, it is better to use words as 'comprehensive' and/or 'systematic'.
- Mention of various cellular processes and the involvement of the RBPs however the authors do not clarify why this is relevant. For example, why is survival in acidic conditions important in the stationary phase "YfiF also binds to csrB and arrS ncRNAs during the stationary phase, two ncRNAs which have a role in cell survival in acidic conditions".
- 'This confirms that integrating total protein abundance and RNA-binding capacity allows...'. This has been done before and I think those studies should be cited as their conclusions are in lines with the conclusions here. Also 'confirms' should be replaced by 'reinforces'.
- In the text, the authors routinely mention "increased RNA binding" without specification on whether this refers to increased binding independent of protein abundance. I think authors should clarify the terminology when referring to the different proteins groups.
- Fig 3c. Not clear what opaque regions are being shown.
- Fig 3f. Unclear what is being aligned. Clarification of figure legend.
- 'Moonlighting RBPs'16. I think this is not the best reference for this statement as it relates to a protein-protein interaction study, however, there is substantial literature of cellular proteins that moonlight as RBPs.
- Fig 4a. Very difficult to read, too small. Clearer indication of Proteobacteria within the tree of life might be useful to readers.
- Fig S5.a Did not write what the meaning of the colours are. (written in same figure b. but not in a.)
- Fig 5.b. What is in orange and grey? More annotation on what the meaning of the volcano plot will be helpful to the reader, left side of volcano vs right side. Different colours for significantly increased and significantly decreased binding. Are there only 2 canonical RBPs in the dataset? if not, then legend should indicate that only CspA and Rml are highlighted.
- Fig 5.e. Clarify figure showing "Aligned reads" from the CLIP dataset.

Reviewer #2:

In this manuscript, Monti et al. present a study analysing the RNA-protein interactome in *E. coli*, with a particular focus on the dynamic changes in these interactions through the different growth stages of this bacterium. The study is technically sound, using methods previously developed in the group, the paper is well written, and the data are interesting and relevant to the community with respect to proteins whose interaction with RNA was not previously known, or where evidence was provided that these interactions depend on growth stage. At the same time, the paper is rather superficial when it comes to determining the functionality of protein-RNA interactions: the authors describe possible roles of RNA binding in bacterial infection or human pathology, and suggest that this could be used as new antibiotic drug targets. However, all of this is rather speculative, as no evidence is provided to support these ideas. Therefore, the main merit of the paper is the provision of an RNA-protein interactome catalogue, with the particular strength that the dynamics of these interactions have been determined across bacterial growth stages - going a step further than most other studies. However, it is a bit of a leap of faith to recognise the functional implications of this work, as the authors attempt to do.

Apart from this general concern that the manuscript may not reach the depth or provide the new insights to make it a strong candidate for this journal, I have a few comments - mostly minor:

1. P5: 'proteins upregulated in the exponential growth phase (Groups 2 and 3) are implicated in metabolic processes which are required for cellular adaptation to robust growth'. This is actually impossible to see from the figure. GO-terms are not always easy to interpret as umbrella terms, however this is made impossible by cutting them off after the first 2 words as done in Fig 1e. In addition, most terms seem to be enriched not only in group 2 and 3 but even more prominently in group 6. Any explanation?
2. P7: '60% of the GO-annotated RBPs decrease in both total abundance and RNA binding in the stationary phase'. It is unclear why this is 'as expected'?
3. P8: their most significant GO-term was 'catalytic activity'. Catalytic activity is a broad term, and the authors provide only one of

several other plausible interpretations by implying a role for RNA binding. For instance, proteins with 'catalytic activity' can bind various other co-factors. To substantiate the authors' assumption, it will be useful to provide more details for the individual proteins in this category, e.g. indicating which/how many actually are known bind nucleotides.

4. P8: 'Out of the 66 proteins that change RNA binding irrespectively of protein amount....' The term 'irrespective' (here and elsewhere in the manuscript) is somewhat ambiguous since it includes the previous category in Q1 and 3. Different wording would be preferred, e.g. changes in RNA binding and protein abundance show anti-correlation. 'Independent' (as used on the same page) is not an ideal term either, since the underlying mechanism of the observed trend in RNA-protein interaction is not known (possibly it is determined by a PTM of CsrA that hinders RNA binding in lag phase. i.e. it is actually dependent on the protein).

5. On page 12-13, authors discuss the conservation of RBPs and what this may mean e.g. for human diseases. These are nice and potentially intriguing observations, however at this stage they are not more than 'circumstantial observations' without concrete or causal evidence. Therefore, without further validation, this section seems better placed in the discussion than in the results section.

6. P16: 'This has led to the characterisation of YfiF as a tRNA binding protein required for bacterial growth.' In fact the experiment showed that YfiF itself is required for bacterial growth, but not that this depends on its RNA-binding capacity.

7. Have the authors investigated if they can derive direct evidence of RNA binding by identifying RNA crosslink sites (e.g. see work by the Urlaub lab)?

8. P12: reference to Extended data fig S5b should be S6b.

9. Header in suppl table S4: data are represented in Fig. 4b, not 5b as indicated.

Reviewer #3:

In the manuscript Monti and colleagues present the first dynamic characterization of RNA-protein interactions throughout different stages in the population growth of bacteria. The authors use a modified version of OOPS combined with mass spectrometry to describe about 90 proteins previously not described to interact with RNA. Comparison to total protein numbers, interestingly revealed that the change in RNA binding is irrespectively of protein amount between stationary and exponential phases. Monti and coworkers use knockouts to show the functional relevance of RBPs on bacterial growth.

This is a well conducted study with thoroughly presented findings. The discovery of novel RBPs in a the well-studied bacterium, E.coli, is of greater general interest and highlights the importance of RBPs on the regulation of bacterial growth.

Minor issue:

The authors describe binding of YfiF to rRNA and tRNAs. Is this binding specific to distinct regions or structures in rRNA and tRNA. Since iCLIP provides nucleotide resolution of crosslinking sites, did the authors find or identify any binding preference for YfiF.

Reviewer #1:

Monti et al. apply here OOPS to study the RBPome of bacteria and the changes that it undergoes during the different cell growth phases. OOPs is very relevant to prokaryotes, because in contrast to eukaryotic mRNAs, bacteria are not polyadenylated and are, therefore, not readily accessible to oligo(dT)-based approaches. It is nice though to see that the presented data overlap very well with previous RBPomes in bacteria while providing substantial newly identified RBPs. Moreover, they study the dynamics of these RBPs during the different growth phases showing that the bacterial RBPome is highly dynamic, as previously shown in mammalian cells. The authors define a group of novel RBPs that are differentially regulated during the bacterial role and provide evidence of binding to several RNAs. The author team is composed of experts in the field, and as such the paper is well written and the data seems solid. There are several points that can be improved and will strengthen this work.

We would like to thank the reviewer for their positive and constructive comments. Please see a detailed response to their suggestions for revision below:

Major points:

- As a general comment, I think adding page numbers and lane numbers would help a lot in the revision of the manuscript.

Page and lane numbers have now been added.

- The introduction does not read as an introduction and indeed the first paragraph is the only one that provides context to the work. The rest is a repetition of the experiments done in this work. I suggest that this must be better balanced expanding the knowledge on the field either from the RNA metabolism in bacteria angle or by explaining how other studies have profiled the dynamics of the eukaryotic (but not prokaryotic) interactome. Either angle would add value to the work by contextualising it better.

As suggested, the introduction has been expanded to include some seminal work on the application of dynamic RBPome studies in eukaryotic systems.

- Authors do not explore one of the main limitations and source of bias of their approach, which is the composition of the transcriptome they purify from bacteria. In order to understand the sources that contribute to the RBPome, it is important to define what the relative abundance of each RNA biotype is in the pull down. An RNAseq analysis of the eluates considering multiple mapping genes (e.g., rRNA) would help to understand the composition of the isolated transcriptome in bacteria and the relative contribution of each RNA biotype to the final results.

We appreciate this is an important consideration in this study and all OOPS-based explorations. Indeed, as previously shown in (PMID: 30607034), where we analysed the transcriptome composition retrieved by OOPS, one of the main strengths of our method is that there is no bias in the recovery of protein-bound transcripts. This is due to the fact OOPS retrieves RNA-protein adducts based on their common physicochemical properties and not on any specific RNA sequence. This has been clarified in the Discussion as follows:

‘As previously shown, the recovery of RNA biotypes bound by RBPs via OOPS is unbiased for transcripts over 60 bp (PMID: 30607034)’.

- The number of replicates in the different experiments appears to be inconsistent or potentially mislabelled. In Fig 1c it is indicated that there were 3 biological replicates, but the figure shows 5 columns in each growth stage, suggesting 5 replicates rather than 3. Maybe I have interpreted something wrongly, but I think this needs further clarification.

Thank you for pointing out this inconsistency. Indeed, the legend was incorrectly stating that there were three replicas. The reviewer is correct and there were indeed five. This error has now been amended.

- In addition, Fig 5.f-g shows 1 control replicate per condition and 2 biological replicates for the samples. If that is the experimental design and is not misinterpreted by me, then the number of replicates in that experiment is not statistically robust enough.

Regarding the replicates in the CLIP analysis in Fig. 5, the peak calling algorithms included in the iCLIP pipeline (<https://app.flow.bio/>) define high confidence peaks in each replicate. In order to identify the binding sites of YfiF and HtpG we have intersected the replicates and blacklisted the peaks identified in the negative controls as these are ‘contaminants’. Therefore, we have not applied an enrichment test over the controls. Instead, we have restricted our analysis on peaks found only in the CL replicates. Non-crosslinked and untagged controls are included to confirm that they have fewer reads/less signals on the gel (and to identify contaminating peaks). All replicates have been included separately in the target plots (data visualised with <https://github.com/ulelab/clipplotr>) to show consistency.

- Bona fide RBPs have physicochemical properties that make them suitable for RNA binding. This is particularly easy to test in bacteria, as proteins are in average smaller than their eukaryotic counterparts and are more globular. Authors should examine general properties such as hydrophobicity (in RBPs because of the accumulation of basic residues) and enrichment for amino acids such as K, R, Y, and others frequently found at protein-RNA interfaces. There have been many studies looking at these amino acid biases in detail. Authors can analyse these parameters for the total proteome, the RBPome, the previously known RBPs and the newly identified RBP group. Finding similar patterns would suggest that these groups are all similar biochemically and thus compatible with RNA binding. This would increase the value of the dataset by

benchmarking it to known features of RBPs. If AlphaFold or crystal structures exist, it could be possible to use software such as BindUP that predict RNA-binding surfaces because of their physicochemical properties. Such an approach could be applied to their candidates (discussed in the last part of the paper) to test if there is a RNA-binding compatible surface and if so, where.

As suggested by the reviewer, we have now examined their hydrophobicity and charge. Furthermore, we have assessed the over-representation of KYR and, as a control, we have also assessed the representation of DNQZ, reported to be depleted in RNA-protein interfaces (Kramer *et al*, 2014). Indeed, comparing RBPs detected by OOPS with the rest of the proteome, we see a statistically significant difference showing RBPs to be more hydrophilic, more charged, enriched in KYR and showing no statistical difference for the negative control (i.e. DNQZ abundance).

The results section has been updated with reference to this analysis, data rendered in Extended Data Fig. S3 as well as a full description of the approach included in the methods.

The BindUP software is currently unavailable and we have not been able to access it after contacting the authors. However, it is worth noting that previous versions had no option for prediction with AlphaFold, so it could only be applied to proteins with solved structures.

- Authors identified 3 RBPs with conserved orthologs between E.coli and H.sapiens including YhgF. It is mentioned the the lack of characterisation of YghF; however, a recent paper showed through its binding profile (CLIP-seq) its potential involvement in transcriptional regulation. I feel expanding the text and cross-referencing to the available CLIP would solidify their results (if in agreement). Also, do the orthologs of these proteins in humans bind to RNA? Do they have CLIP-seq datasets? How do they compare to bacterial data? What do we know about them that can be used to understand the bacterial counterpart.

As suggested, we have now expanded the section on YhgF to include more details on the recent CLIP study as follows:

“While annotations for YhgF are very limited and the function of the human ortholog (SRBD1) is unclear, the interactome of this protein has been recently characterised in bacteria and was found to bind both mRNAs, tRNAs and sRNAs. Indeed, overexpression of yhgF leads to increased levels of its target rnf mRNA indicating a role in transcriptional regulation.”

As indicated in the manuscript (page 12), all three human orthologues have been identified as RBPs. However, no CLIP data for any of the orthologs is available to our knowledge.

- Authors use a STRING-db prediction for YfiF and HtpG interaction networks. Are there studies determining it experimentally? What is known of the partners of the cellular orthologs?

To build the interaction networks, proteins were queried against the *Escherichia coli* K12 substr. MG1655 pre-loaded database on Cytoscape (v 3.9.1) and filtered to keep only experimentally determined physical interactions.

YfiF does not have orthologs in humans. With regards to HtpG, the direct ortholog is Heat Shock protein HSP90-alpha (HSP90AA1, P07900) which has been identified as RNA-binding in two publications (Baltz *et al*, 2012; Castello *et al*, 2012). To our knowledge there is no specific study completed on the protein interactome of HSP90AA1. The interactors listed in the STRING database are shown below, where only physical interactors, with a confidence level > 0.4 are shown).

Figure 1: Physical interaction network of HSP90AA1, human ortholog of HtpG, as annotated in STRING-db.

These interactors include CDC37, a co-chaperone of HSP90AA1. In the introduction, we noted that “P23, a cochaperone of HSP90, [is] now validated to interact with mRNAs” in macrophages; suggesting perhaps a similar role for CDC37. Furthermore, another molecular chaperone HSPA8 (Heat shock cognate 71 kDa protein) annotated as RNA binding is found as an interactor (Castello *et al*, 2012; Baltz *et al*, 2012).

- Further characterization of the CLIPs of newly discovered proteins would improve the value of this work. The authors show binding sites on the overall cellular transcriptome and maps of a few selected transcripts. I feel this CLIP analysis is treated very superficially. Which binding sites are statistically enriched? Do these proteins recognize any specific sequence? Do they bind any signature within the target RNAs? For example, for the tRNAs, do they recognize a particular stem-loop? Plots showing binding site density across the different nucleotides of the tRNAs could help to define the binding signature of the protein. It seems to me that the CLIP-seq work is

underexplored and after putting so much effort into it, I feel it should be analysed deeper.

We thank the reviewer for analysis suggestions. In order to explore the data in more depth, as well as apply the same pipeline to both yfiF and htpG targets, the demultiplexed target reads were uploaded onto <https://flow.bio/> and the CLIP-seq v1.0 pipeline was run. From the resulting peaks, all peaks intersecting with those found in the negative sample (no tagged protein) were blacklisted and removed. Furthermore, in order to keep a high confidence list we kept the peaks that were not identified in the respective NC samples as well. Statistical tests can not be applied to define peaks of interest as the optimal peak will have no signal from the controls. Therefore, we have only included the negative controls as visual markers for the gel as well as for our CLIP plots rather than for statistical purposes. Instead, we selected peaks to highlight by their score as well as standard deviation between the replicates as well as by functional data integration. Two of these were short ncRNA (arrS and chiX) for which we could obtain a robust RNA folding prediction with ViennaFold and have included.

Motifs were explored using STREME (Bailey, 2021) where the CL peaks were analysed using the NC as negative controls and the top motif was reported as well as PEKA for k-mer enrichments. In the case of HtpG, as anticipated by the reviewer, our analysis shows that this motif is located in the anticodon loop of the tRNA targets and has been included in Fig. 5 and Extended Data Fig. 10. In the case of YfiF, the top motif was found in several rRNA genes. Moreover, AU-rich k-mers were found enriched (Extended Data Fig. 7) in YfiF motifs and shown visually in Fig. 3j.

- Following on the CLIP-seq experiments, the authors used MiSeq sequencers for their CLIP sequencing that produce only a few million reads per lane. Hence, the read depth provided by a MiSeq does not seem sufficient for a comprehensive CLIP analysis. In other bacterial CLIP experiments, Illumina NextSeq sequencers have been used. Could the authors specify how many samples were run per run and what read depth they accomplished? Are these sufficient for robust statistical analyses?

Eight samples were run as follows:

One negative control, no his-tag in genome and crosslinked;

One htpG negative control, htpG-his-tag E. coli non-cross-linked;

Two htpG samples, htpG-his-tag E. coli cross-linked;

One yfiF negative control, yfiF-his-tag E. coli non-cross-linked;

Three yfiF samples, yfiF-his-tag E. coli cross-linked.

As mentioned above, hits in either of the negative controls were used for blacklisting. All replicate data has been shown in the figures to indicate that the signal is enriched over the negative.

With regards to the sequencing depth, we have evaluated whether the libraries need re-sequencing by generating the ratio of cDNA (unique reads) over total reads

obtained by the UMI analysis after deduplication at <https://flow.bio> and indeed, the ratio does approximate 1 in the case of the negative controls suggesting a lack of library complexity as expected from these samples. However, in the case of the cross-linked samples this is not the case indicating that the read depth is robust.

Type	NG_CL_R1	YFIF_NC_R1	YFIF_CL_R1	YFIF_CL_R2	YFIF_CL_R3
Reads processed	267028	241102	3058904	2040191	2240161
Unique mapped reads (STAR)	46111	23981	380727	210352	196894
Number of unique alignment positions	16733	11149	52057	40079	35983
Number of reads after deduplicating	45229	23277	311102	169176	138901
UNIQUE READS over DEDUP reads	1.01950076	1.03024445	1.2238012	1.2433915	1.41751319

Figure 2: Number of reads per sample type in the different steps of YfiF iCLIP analysis. Values shown in table above. Error bars show standard deviation between replicates.

Type	NG_CL_R1	HTPG_NC_R1	HTPG_CL_R1	HTPG_CL_R2
Reads processed	267028	250724	1928226	2411435
Unique mapped reads (STAR)	46111	31281	314426	427083

Number of unique alignment positions	16733	13664	37434	42992
Number of reads after deduplicating	45229	30879	246600	280301
UNIQUE READS over DEDUP reads	1.019500763	1.013018556	1.275044607	1.523658496

Figure 3: Number of reads per sample type in the different steps of HtpG iCLIP analysis. Values shown in table above. Error bars show standard deviation between replicates.

Furthermore, in the PNK assay (Extended Data Fig S5 and S10), we observed that the length of the interacting RNAs was short and the expected abundance of the interactors low. Therefore, given the expected limited number of target transcripts and their short length MiSeq provided enough coverage for our analysis.

- Authors showed that ablation of their candidate proteins impair cellular growth, however, it isn't clear if this is due to intrinsic toxicity or impaired cell division. Also, no connection with their RNA binding activities is provided. I think this part can be expanded with a few selected experiments (e.g. do the bacteria look normal under the microscope? Can they divide normally?) or supporting their data with pre-existing research (if available).

As suggested by the reviewer, additional experiments were performed to ascertain whether the lower growth rates in some of the knockouts corresponded to problems in cell division or other morphological alterations. Phase contrast imaging and automated image analysis, showed that in both the richer (M9LB) and the poorer medium (M9+1% glycerol), the knockouts had no significant morphological changes, with no signs of filamentation or division impairment. This is now reported in the main text on Page 10: 'we analysed the effects of knocking out these proteins and evaluated

the growth phenotype and morphology in rich and growth-limiting defined media', with the data presented in a new Extended Data Figure S4.

Importantly, the KO library was intended as a first selection step to detect candidates altering cell growth for further validation, and not to directly determine a link between RNA binding and growth.

- Fig 5.f-g. The authors do not describe or mention either of these figures in the text.

We have now updated Figure 5 with the extended CLIP analysis.

MINOR POINTS:

- Abstract: 'provide the first dynamic RBPome'; I don't think this is correct > 'for the first time the dynamics of the bacterial RBPome'

This has been amended.

- 'This unbiased method' > I would recommend authors to avoid this assumption as the method is biased. First, UV crosslinking would favour proteins that bind to single stranded regions, second, mRNAs are a small subpopulation in the context of total RNA. Most methods are biased, and, therefore, it is better to use words as 'comprehensive' and/or 'systematic'.

When we say 'unbiased method' we are alluding to the fact that OOPS captured RBP bound to all types of RNA, not just mRNA as previous methods. But we appreciate this is not clear- so we have included this in more detail in the introduction. Moreover, we have changed this statement to 'comprehensive' as suggested by the reviewer.

- Mention of various cellular processes and the involvement of the RBPs however the authors do not clarify why this is relevant. For example, why is survival in acidic conditions important in the stationary phase "YfiF also binds to csrB and arrS ncRNAs during the stationary phase, two ncRNAs which have a role in cell survival in acidic conditions' '.

As the cells were grown in batch culture (closed culture system) and the M9LB media contains glucose, we expect a gradual acidification of the media as the cells consume the available glucose(Walczak *et al*, 2023). We acknowledge this was not clear in the text and have added the following:

'As the cells were cultured in M9LB which contains glucose, we expect fermentation and acidification of the media through-out batch culture, suggesting a putative upregulation of these ncRNAs.'

- 'This confirms that integrating total protein abundance and RNA-binding capacity allows...'. This has been done before and I think those studies should be cited as their conclusions are in lines with the conclusions here. Also 'confirms' should be replaced by 'reinforces'.

The text has been changed to 'reinforces' and the references to the following papers that have incorporated total protein abundance in their analysis/interpretation have been cited:

Perez-Perri, J.I., Ferring-Appel, D., Huppertz, I. *et al.* The RNA-binding protein landscapes differ between mammalian organs and cultured cells. *Nat Commun* 14, 2074 (2023). <https://doi.org/10.1038/s41467-023-37494-w>

Queiroz, R.M.L., Smith, T., Villanueva, E. *et al.* Comprehensive identification of RNA–protein interactions in any organism using orthogonal organic phase separation (OOPS). *Nat Biotechnol* 37, 169–178 (2019). <https://doi.org/10.1038/s41587-018-0001-2>

Sysoev VO, Fischer B, Frese CK, Gupta I, Krijgsveld J, Hentze MW, Castello A, Ephrussi A. Global changes of the RNA-bound proteome during the maternal-to-zygotic transition in *Drosophila*. *Nat Commun*. 2016 Jul 5;7:12128. doi: 10.1038/ncomms12128. PMID: 27378189; PMCID: PMC4935972.

- In the text, the authors routinely mention "increased RNA binding" without specification on whether this refers to increased binding independent of protein abundance. I think authors should clarify the terminology when referring to the different protein groups.

We apologise for the lack of clarity. The text has been clarified as follows:

“Out of the 66 proteins where RNA binding is not correlated to protein amount between stationary and exponential phases (FDR < 0.01), only 24 were annotated as RBPs”

And

“All three displayed increased RNA binding in the stationary phase alongside a concomitant increase in protein abundance (Q1, Fig. 2c), showing consistent binding profiles across the phases (Fig. 5b).

“

- Fig 3c. Not clear what opaque regions are being shown.

To facilitate the distinction between the 2 highlighted domains (i.e. SpoU-like MTase and the substrate binding domain) the figure has been re-rendered and different colours used.

- Fig 3f. Unclear what is being aligned. Clarification of figure legend.

This figure has been updated with the revised CLIP-analysis. It now indicates the unique crosslinks (i.e. cDNA molecules) identified in the YfiF iCLIP data-sets summarised by gene type, this is produced by the 'ICOUNT_SUMMARY' process of the iCLIP pipeline used as detailed above. Note that pre-mapped refers to the standard bowtie2 pre-mapping step to highly redundant sequences to remove multi-mapping reads and that intergenic includes tRNA and rRNA sequences. This has been made more explicit in the legend as recommended by the reviewer.

- 'Moonlighting RBPs'¹⁶. I think this is not the best reference for this statement as it relates to a protein-protein interaction study, however, there is substantial literature of cellular proteins that moonlight as RBPs.

We thank the reviewer for this comment. The citation has been modified to include the following reviews:

Castello, A., Hentze, M.W. and Preiss, T. (2015) Metabolic enzymes enjoying new partnerships as RNA-binding proteins. *Trends Endocrinol. Metab.* 26, 746–757

Hentze, M.W., Castello, A., Schwarzl, T. and Preiss, T. (2018) A brave new world of RNA-binding proteins. *Nat. Rev. Mol. Cell Biol.* 19, 327–341

Cieśla, J. (2006) Metabolic enzymes that bind RNA: yet another level of cellular regulatory network? *Acta Biochim. Pol.* 53, 11–32

Nicole J. Curtis, Constance J. Jeffery; The expanding world of metabolic enzymes moonlighting as RNA binding proteins. *Biochem Soc Trans* 30 June 2021; 49 (3): 1099–1108

- Fig 4a. Very difficult to read, too small. Clearer indication of Proteobacteria within the tree of life might be useful to readers.

The purpose of the plot in 4A is to show the loss of conservation of the functionally uncharacterised RBPs as the distance to *E. coli* increases. Given they are mainly conserved in the branch of Proteobacteria this has been expanded in 4B and the table of results included in Dataset EV5 and the legend has been edited to redirect readers to said table.

- Fig S5.a Did not write what the meaning of the colours are. (written in the same figure b. but not in a.)

This figure is now Extended Data Figure 8. The clarification of the colours has been amended in the legend of S8a, moreover in legend S8b we have clarified that only YhgF is plotted.

- Fig 5.b. What is in orange and grey? More annotation on what the meaning of the volcano plot will be helpful to the reader, left side of volcano vs right side.

Different colours for significantly increased and significantly decreased binding. Are there only 2 canonical RBPs in the dataset? if not, then legend should indicate that only CspA and Rml are highlighted.

The legend has been modified to indicate that only CspA and Rmf are highlighted. Further, the plots have been re-generated to incorporate the reviewers comments: orange now labelled as 'sig. Increase RNA binding' while sky blue indicates 'sig. Decrease RNA binding'. Furthermore, the plot titles have been updated to help guide the reader in understanding which phases are being compared. Thank you for the observation and we hope it is clearer now.

- Fig 5.e. Clarify figure showing "Aligned reads" from the CLIP dataset.

This figure has been updated with the revised CLIP-analysis- it now indicates the unique crosslinks (i.e. cDNA molecules) identified in the HtpG iCLIP data-sets summarised by gene type as indicated above in the case of Fig. 3f. As above, the description of the figure has been made more explicit in the legend as recommended by the reviewer.

Reviewer #2:

In this manuscript, Monti et al. present a study analysing the RNA-protein interactome in *E. coli*, with a particular focus on the dynamic changes in these interactions through the different growth stages of this bacterium. The study is technically sound, using methods previously developed in the group, the paper is well written, and the data are interesting and relevant to the community with respect to proteins whose interaction with RNA was not previously known, or where evidence was provided that these interactions depend on growth stage. At the same time, the paper is rather superficial when it comes to determining the functionality of protein-RNA interactions: the authors describe possible roles of RNA binding in bacterial infection or human pathology, and suggest that this could be used as new antibiotic drug targets. However, all of this is rather speculative, as no evidence is provided to support these ideas. Therefore, the main merit of the paper is the provision of an RNA-protein interactome catalogue, with the particular strength that the dynamics of these interactions have been determined across bacterial growth stages - going a step further than most other studies. However, it is a bit of a leap of faith to recognise the functional implications of this work, as the authors attempt to do.

Apart from this general concern that the manuscript may not reach the depth or provide the new insights to make it a strong candidate for this journal, I have a few comments - mostly minor:

1. P5: 'proteins upregulated in the exponential growth phase (Groups 2 and 3) are implicated in metabolic processes which are required for cellular adaptation to robust growth'. This is actually impossible to see from the figure. GO-terms are not always easy to interpret as umbrella terms, however this is made impossible by cutting them off after the first 2 words as done in Fig 1e. In addition, most terms seem to be enriched not only in group 2 and 3 but even more prominently in group 6. Any explanation?

Following the reviewer's comments we have now updated the GO term plot and have now extended the x-axis labels for ease of interpretation.

GO-term enrichment assumes complete and robust annotation of all proteins, in an equal manner. However, we acknowledge that GO terms are not exhaustively annotated. If we pull all the annotations available for each of the groups we see there is more information for categories 1 and 6 (see figure below), which are also the categories with more significant P-values (please note that size is inversely proportional to P-value). This is directly dependent on these sets having more proteins than the others. Considering this, the plot is rendered such that at least 10 GO terms per category are represented. As group 6 overlaps with several terms from other categories, there are more points than in the rest. To clarify this, we have now included a new table 'Dataset EV2' including all available GO term annotations of the identified proteins.

Figure 4: Count of unique GO-terms associated to protein sets as grouped in Figure 1 of the manuscript. Total number indicated above each bar.

- P7: '60% of the GO-annotated RBPs decrease in both total abundance and RNA binding in the stationary phase'. It is unclear why this is 'as expected'?

We appreciate this remark may not have been clear enough and have now made a more explicit comment in the text. This statement was related to the observation that ~90 proteins of the 180 GO-annotated RBPs are related to translation (Holmqvist & Vogel, 2018), e.g. ribosomal proteins, tRNA synthetases, translation factors, etc. As translation is known to decrease in the stationary phase, as shown in (Reier *et al*, 2023), these proteins are expected to decrease their RNA binding activity.

- P8: their most significant GO-term was 'catalytic activity'. Catalytic activity is a broad term, and the authors provide only one of several other plausible interpretations by implying a role for RNA binding. For instance, proteins with 'catalytic activity' can bind various other cofactors. To substantiate the author's assumption, it will be useful to provide more details for the individual proteins in this category, e.g. indicating which/how many actually are known to bind nucleotides.

We have now included a new Dataset EV2 with all proteins enriched per GO term. We have observed that 35% of the proteins with GO annotated catalytic activity are currently annotated as nucleotide binding.

Figure 5: Pairwise venn diagram representation of proteins annotated as ‘Catalytic binding’ (GO:0003824) and/or ‘Nucleotide binding’ (GO:0000166) for total proteome (above) and proteins belonging to Category 1 (below).

4. P8: ‘Out of the 66 proteins that change RNA binding irrespective of protein amount....’ The term ‘irrespective’ (here and elsewhere in the manuscript) is somewhat ambiguous since it includes the previous category in Q1 and 3. Different wording would be preferred, e.g. changes in RNA binding and protein abundance show anti-correlation. ‘Independent’ (as used on the same page) is not an ideal term either, since the underlying mechanism of the observed trend in RNA-protein interaction is not known (possibly it is determined by a PTM of CsrA that hinders RNA binding in lag phase. i.e. it is actually dependent on the protein).

We agree with the reviewer that the wording may not have been clear enough. We have now amended the text as follows: ‘Out of the 66 proteins where RNA binding is not correlated to protein amount (Q2 and Q4)....’

We agree that the use of ‘independent’ as was previously worded could have led to confusion, this has been amended to: “In keeping with these findings, we found a

significant increase in RNA-binding of CsrA (P -value: 1.92E-7) in the stationary phase, that is independent of the total abundance CsrA protein (Fig. 2d)”

5. On page 12-13, authors discuss the conservation of RBPs and what this may mean e.g. for human diseases. These are nice and potentially intriguing observations, however at this stage they are not more than ‘circumstantial observations’ without concrete or causal evidence. Therefore, without further validation, this section seems better placed in the discussion than in the results section.

Although we agree further work on the evolutionary conservation of these RBPs would be of interest and have specified this in the new version of the manuscript’s discussion, we believe pointing to a link to disease related mitochondrial RBPs provides useful information for result interpretation.

6. P16: ‘This has led to the characterisation of YfiF as a tRNA binding protein required for bacterial growth.’ In fact the experiment showed that YfiF itself is required for bacterial growth, but not that this depends on its RNA-binding capacity.

Following the reviewer’s comment, we have revised the claim to better represent our results: ‘This has led to the characterisation of YfiF as a tRNA binding protein as well as a protein required for bacterial growth.’

7. Have the authors investigated if they can derive direct evidence of RNA binding by identifying RNA crosslink sites (e.g. see work by the Urlaub lab)?

We thank the reviewer for this suggestion. However, it is important to note that our study was designed with a focus on quantification of RNA binding dynamics and not on the detection of specific RNA crosslink sites, which is not compatible with quantitation. Nevertheless, we have contacted the Urlaub lab and run the samples using their newest computational tool named NuXL, that is used to detect the sites of RNA crosslinking to specific amino acids in RBPs. Since the samples included in the manuscript are not enriched for crosslinked peptides, we obtained no signal when using NuXL, as the abundance of linear peptides (i.e. non-crosslinked peptides coming from the same RBP) in the mixture suppresses the signal from the crosslinks.

As a proof of concept, we proceeded to prepare samples enriching for RNA-protein crosslinks by TiO₂ as described in the Methods of (Queiroz *et al*, 2019). Indeed, the signal was improved but the results were biased towards proteins of high abundance (see figure below), with over half of the ~30 proteins detected being ribosomal.

Figure 6: Protein abundance values (as determined by TMT experiment) of proteins classified by presence in bonafide RBPome ('sig') and identified by NuXL ('RELIABLE') or not.

In total, only 31 proteins were detected with high reliability, of which 26 were detected as RBPs in our study (see figure)-. The remaining 5 were not identified.

Intersection between NuXL and RNase

Figure 7: Stacked bar plot of intersection between predicted peptide RNA-binding sites in RBPs with the RNase assay defining the bonafide RBPome. RELIABLE: high confidence identification of RNA-bound peptide; LOW-LOC: low localisation value, spectra must be interpreted manually to confirm if true positive; ENR-FP: enriched false positives, identifications of low confidence.

8. P12: reference to Extended data fig S5b should be S6b.

Apologies for the confusion here. This reference to Extended Data Fig. S5b (now S8b) was not well placed and has been removed. It indicates the conserved structure of YhgF between human and E. coli and has been referred to earlier on P12. Extended Data Figure S6b (now S9b) instead refers to the interactome of the ClpB protein.

9. Header in supplementary table S4: data are represented in Fig. 4b, not 5b as indicated.

Thank you for the comment. This has now been corrected. Please note that due to edits to the manuscript this table is now Dataset EV5.

Reviewer #3:

In the manuscript Monti and colleagues present the first dynamic characterization of RNA-protein interactions throughout different stages in the population growth of bacteria. The authors use a modified version of OOPS combined with mass spectrometry to describe about 90 proteins previously not described to interact with RNA. Comparison to total protein numbers, interestingly revealed that the change in RNA binding is irrespective of protein amount between stationary and exponential phases. Monti and coworkers use knockouts to show the functional relevance of RBPs on bacterial growth.

This is a well conducted study with thoroughly presented findings. The discovery of novel RBPs in a well-studied bacterium, E.coli, is of greater general interest and highlights the importance of RBPs on the regulation of bacterial growth.

We would like to thank the reviewer for their positive feedback on our article.

Minor issue:
The authors describe binding of YfiF to rRNA and tRNAs. Is this binding specific to distinct regions or structures in rRNA and tRNA. Since iCLIP provides nucleotide resolution of crosslinking sites, did the authors find or identify any binding preference for YfiF.

The CLIP analysis has been updated to explore the data in more depth. In the case of YfiF, we have found a broad spectrum of interaction partners, apart from its own mRNA (Fig. 3g). We identified the motif **AACCTTTACW** located across multiple rRNA genes, but also across some tRNAs and interestingly, in several of the non-coding RNA target genes (Dataset EV4). In addition, in order to explore this further, we have performed a positionally enriched k-mer analysis (PEKA) (Extended Data Fig. S7). This approach allows identifying enriched kmers from iCLIP datasets by considering the cDNA count as well as the technical biases of UV-crosslinking. In this manner, we have observed that YfiF has a preference for AU-rich sequences. Two of these were present in short ncRNA (arrS and chiX) for which we could obtain a robust RNA folding prediction with ViennaFold and the identified binding sites are indeed located in AU-rich stem-loops.

In the case of HtpG, we have found a clear preference for tRNAs (Fig. 5f), specifically binding in the anticodon loop. No self-binding or binding to non-coding RNAs has been identified.

4th Mar 2024

Manuscript Number: MSB-2023-12032R

Title: Interrogation of RNA-protein interaction dynamics in bacterial growth

Dear Eneko,

Thank you for sending us your revised manuscript. We have now heard back from the reviewer who was asked to evaluate your revised study. As you will see below, reviewer #1 thinks that most of the previously raised concerns have been addressed. However, they list two remaining issues, which we would ask you to address in a revision. We would also ask you to address some editorial issues listed below.

- Our Data Editors noted that the following needs to be corrected/added in the Figure Legends:

-- Figure panel 1e is not labelled in the figure. Please correct.

-- Please provide a legend for figure 7h.

-- Please indicate the statistical test used for data analysis in the legends of figures 1e; 2c, e; 3d; 5a-b.

-- Please include information related to n in the legend of figure 1d.

-- We noted that n=2 in figure 5e. As per our author guidelines, we recommend that the actual individual data from each experiment should be plotted if $n < 5$, alongside an error bar.

-- The measure of center for the error bars needs to be defined in the legends of figures 3b; 3f; 5e.

- The funding information provided in the manuscript text (Acknowledgements) should match the information entered in the online submission system. The information on: Cancer Research UK (FC001110), the UK Medical Research Council (FC001110), and the Wellcome Trust (FC001110); Biotechnology and Biological Sciences Research Council through the Industrial Biotechnology Catalyst grant BB/N01040X/1 is missing from the submission system.

- Please include the 5 keywords in the main text, below the Abstract.

- Please remove the 'Authors Contributions' from the manuscript. The 'Author Contributions' section is replaced by the CRediT contributor roles taxonomy to specify the contributions of each author in the journal submission system. Please use the free text box in the 'author information' section of the online submission system to provide more detailed descriptions if needed (e.g., 'X provided intracellular Ca⁺⁺ measurements in fig Y').

- There is a callout for Fig. 1E and Fig. 5I, but the panels are not labeled. Moreover, a callout for Fig. 4D is missing.

- Please include a description of the EV Dataset, in a separate sheet within the xls for Datasets EV8-EV10.

- Regarding the Appendix file: the nomenclature should be Appendix Figure S1-S10 in the Table of Contents, figure legends and callouts.

Please resubmit your revised manuscript online, with a covering letter listing amendments and responses to each point raised by the referees. Please resubmit the paper ****within one month**** and ideally as soon as possible. If we do not receive the revised manuscript within this time period, the file might be closed and any subsequent resubmission would be treated as a new manuscript. Please use the Manuscript Number (above) in all correspondence.

Click on the link below to submit your revised paper.

Kind regards,

Maria

Maria Polychronidou, PhD
Senior Editor
Molecular Systems Biology

If you do choose to resubmit, please click on the link below to submit the revision online before 3rd Apr 2024.

IMPORTANT:

Please note that corresponding authors are required to supply an ORCID ID for their name upon submission of a revised manuscript (EMBO Press signed a joint statement to encourage ORCID adoption).

(<https://www.embopress.org/page/journal/17444292/authorguide#editorialprocess>)

Currently, our records indicate that the ORCID for your account is 0000-0002-3585-8846.

Link Not Available

***** PLEASE NOTE ***** As part of the EMBO Press transparent editorial process initiative (see our Editorial at <https://dx.doi.org/10.1038/msb.2010.72> , Molecular Systems Biology will publish online a Review Process File to accompany accepted manuscripts. When preparing your letter of response, please be aware that in the event of acceptance, your cover letter/point-by-point document will be included as part of this File, which will be available to the scientific community. More information about this initiative is available in our Instructions to Authors. If you have any questions about this initiative, please contact the editorial office (msb@embo.org).

Reviewer #1:

Authors have answered most of my comments satisfactorily with two main exceptions:

1. The RBPome can be biased based on RNA biotype abundance. To determine to what extent each RNA specie is contributing to the proteomic results it would be important to know the distribution of the different RNA biotypes in the samples. For example, if tRNAs accounted for 30% of the reads, we would expect tRNA-binding proteins to be substantially represented in the dataset. I still think that knowing the proportion of RNA biotypes in total RNA isolation (using theirs or available data) would be helpful.
2. CLIP is a temperamental method in which a mistake in one library preparation can lead to near no reads and due to the intrinsic nature of the method (and lack of intermediate QC) is difficult to notice or narrow down the problem. This is why having a single negative control is dangerous as the lack of reads can be derived from a particular sample prep. This is why it is critical to keep in mind this when speaking about the results and even if using qualitative means (peak vs no peak), acknowledge the limitations of the experimental setting. A way to get around it is to compare between "positive" examples. As RBPs are not expected to bind to the same regions (unless they are involved in the same pathways), it could be possible to use the negative samples and the signal from the other RBPs to show the specificity of the RBP of interest.

Reviewer #1:

Authors have answered most of my comments satisfactorily with two main exceptions:

1. The RBPome can be biased based on RNA biotype abundance. To determine to what extent each RNA specie is contributing to the proteomic results it would be important to know the distribution of the different RNA biotypes in the samples. For example, if tRNAs accounted for 30% of the reads, we would expect tRNA-binding proteins to be substantially represented in the dataset. I still think that knowing the proportion of RNA biotypes in total RNA isolation (using theirs or available data) would be helpful.

We agree with the reviewer that knowing the distribution of the different RNA biotypes is critical to interpret the results. This is the reason why the OOPS method was chosen to perform this RBPome study. As we proved in our previous work, the relative proportion of each RNA species in OOPS is the same one you can obtain using a total RNA extraction. Please see data presented in Figure 1c,d in the original OOPS manuscript: <https://www.ncbi.nlm.nih.gov/pmc/articles/PMC6591131/figure/F1/>.

The only difference we found in the relative abundance of RNAs captured using this methodology is an underrepresentation of small RNAs (mostly tRNAs). We speculated that since tRNAs diffuse freely in the cell, they may not be in constant contact with RBPs. Moreover, since the number of simultaneous interactions that small RNAs can establish with RBPs may be smaller, the probability of stabilising RNA-protein interactions in short RNAs could be lower. We have previously observed this in all biological systems interrogated, including *E. coli*, see supplementary information in (Villanueva E., *et al*, Nat Protoc. 2020).

Even with this underrepresentation of small RNAs in OOPS (whether technical or biological) we have found YfiF and HtpG interacting with small RNAs, pointing to an interaction which could potentially be even greater *in vivo*.

To clarify this point, we have now included the following text in the discussion:

In this study, we reveal that HtpG may have an extra function as a tRNA binding protein. This is especially relevant considering that small RNA species can be underrepresented in OOPS owing to inefficient recovery during aqueous:organic phase separation (Queiroz *et al*, 2019). Importantly, HtpG interacts with [...]

2. CLIP is a temperamental method in which a mistake in one library preparation can lead to near no reads and due to the intrinsic nature of the method (and lack of intermediate QC) is difficult to notice or narrow down the problem. This is why having a single negative control is dangerous as the lack of reads can be derived from a particular sample prep. This is why it is critical to keep in mind this when speaking about the results and even if using qualitative means (peak vs no peak), acknowledge the limitations of the experimental setting. A way to get around it is to compare between "positive" examples. As RBPs are not expected to bind to the same regions (unless they are involved in the same pathways), it could be possible to use the negative samples and the signal from the other RBPs to show the specificity of the RBP of interest.

While we agree that CLIP is a method that may require some time to optimise, we have actually included intermediate QC steps that are presented in the Appendix Figure S5 and S10. There, it can be seen that the negative controls (both non-crosslinked and untagged) give minimal signal already at the ligation steps. All ligation steps were performed in parallel; suggesting that the lower signal obtained in the negative controls is not due to a potential issue when ligating, generating, or amplifying the CLIP libraries, but the result of not having a target to amplify.

In principle, we agree with the reviewer that using data from other RBPs could be used to define specificity as shown by (Lorenz DA., *et al*, Nat Methods, 2023). However, this has to be done with caution unless many RBPs have been pulled down simultaneously. Given that the multimapping and fragment length of the resulting libraries are quite different between YfiF and HtpG, it would be challenging to compare them fruitfully. Moreover, as the reviewer rightly points out, in order to compare the peaks, we would need to rule out the possibility that they are involved in related pathways. This is not necessarily the case since both candidate RBPs YfiF and HtpG bind more RNA at the stationary phase, and both show evidence of interaction with ribosomal RNA. Moreover, both RBPs interact with small RNAs including tRNAs (yet their top binding motif is different). Taken together, it not clear if they are completely independent. Further studies would determine if these proteins are part of a common stationary-phase biological post transcriptional program and, while we find it hard to believe that proteins like HtpG, could bind to the same sequence in over 16 different tRNAs by chance, we have followed the reviewer's suggestion and toned down and clarified our observations as follows:

Line 417:

[...] we reveal that HtpG may have an extra function as a tRNA binding protein [...];

Line 34:

[...] Peaks overlapping in both experimental replicates and not in the controls were taken forward and visualised by peak score and size, showing a preference for ncRNAs genes, mostly tRNAs (Fig. 5 e,f, Extended Data S10e)[...]

We have also made clearer the number of negative controls used:

Line 345:

To explore whether there was a particular structure or sequence motif within the tRNAs for HtpG binding, the CL peaks were analysed through STREME (Bailey, 2021) using the NC sample as negative control. The top motif was HGGWTTTYAA (Fig. 5g), identified in 16 of the target tRNAs consistently in the anticodon binding loop (Fig. 5h,i, Dataset EV7).

We have specified that as a result of having one negative control the analysis is qualitative to make the limitations of CLIP-seq analysis more patent in the discussion by adding:

Line 415:

Importantly, our qualitative analysis shows that HtpG interacts with [...]

11th Mar 2024

Manuscript number: MSB-2023-12032RR

Title: Interrogation of RNA-protein interaction dynamics in bacterial growth

Dear Eneko,

Thank you again for sending us your revised manuscript. We are now satisfied with the modifications made and I am pleased to inform you that your paper has been accepted for publication.

Kind regards,

Maria

Maria Polychronidou, PhD
Senior Editor
Molecular Systems Biology
